# Stability and learning in excitatory synapses by nonlinear inhibitory plasticity

Christoph Miehl[1,2]*, Julijana Gjorgjieva[1,2]*

**1** Max Planck Institute for Brain Research, Frankfurt am Main, Germany, **2** School of Life Sciences, Technical University of Munich, Freising, Germany

* christoph.miehl@brain.mpg.de (CM); gjorgjieva@tum.de (JG)

## Abstract

Synaptic changes are hypothesized to underlie learning and memory formation in the brain. But Hebbian synaptic plasticity of excitatory synapses on its own is unstable, leading to either unlimited growth of synaptic strengths or silencing of neuronal activity without additional homeostatic mechanisms. To control excitatory synaptic strengths, we propose a novel form of synaptic plasticity at inhibitory synapses. Using computational modeling, we suggest two key features of inhibitory plasticity, dominance of inhibition over excitation and a nonlinear dependence on the firing rate of postsynaptic excitatory neurons whereby inhibitory synaptic strengths change with the same sign (potentiate or depress) as excitatory synaptic strengths. We demonstrate that the stable synaptic strengths realized by this novel inhibitory plasticity model affects excitatory/inhibitory weight ratios in agreement with experimental results. Applying a disinhibitory signal can gate plasticity and lead to the generation of receptive fields and strong bidirectional connectivity in a recurrent network. Hence, a novel form of nonlinear inhibitory plasticity can simultaneously stabilize excitatory synaptic strengths and enable learning upon disinhibition.

**Data Availability Statement:** All relevant data are within the manuscript and its Supporting information files. All code is available at https://github.com/comp-neural-circuits/Nonlinear-inhibitory-plasticity.

## Author summary

An important task the brain needs to solve is the so-called 'stability-flexibility problem'. On the one hand, any representation in the brain, for example a long-lasting memory, has to be stable for a long time. On the other hand, new representations need to be flexibly learned at any time. Learning and memory formation are implemented through the plasticity of synaptic connections, which describe how the activity in neurons is translated into changes of synaptic strength between these neurons. We propose a novel form of synaptic plasticity at synapses from inhibitory to excitatory neurons as a mechanism to stabilize learned representations, while a gating signal triggers the learning of new representations. We identify the dominance of inhibition over excitation and a nonlinear dependence of inhibitory plasticity on the postsynaptic firing rate as important aspects of our newly proposed plasticity mechanism. Our computational model allows us to uncover the underlying mechanism behind various experimental findings related to synaptic plasticity and sensory perturbations, and we formulate multiple experimentally-testable predictions.

**Funding:** CM and JG thank the Max Planck Society for funding and a NARSAD Young Investigator Grant from the Brain and Behavior Research Foundation to JG. We also thank the Deutsche Forschungsgemeinschaft (DFG) for funding through the Collaborative Research Centre (CRC) 1080. The funders had no role in study design, data collection and analysis, decision to publish, or preparation of the manuscript.

**Competing interests:** The authors have declared that no competing interests exist.

# Introduction

Learning and memory formation in the brain are hypothesized to be implemented by synaptic changes undergoing Hebbian plasticity whereby joint pre- and postsynaptic activity increase the strength of synaptic connections [1, 2]. However, Hebbian long-term plasticity of excitatory synapses to other excitatory neurons, referred to as excitatory plasticity, is inherently unstable [3]. Increasing excitatory synaptic strengths leads to an increase in the firing rates of excitatory postsynaptic neurons which in turn further increases synaptic strengths. This positive feedback loop is called 'Hebbian runaway dynamics' [4]. To counteract unstable synaptic growth and control resultant rate dynamics, some form of homeostatic control is needed. Experimental studies have uncovered multiple homeostatic mechanisms. One prominent mechanism is synaptic scaling, where synaptic connections onto a given excitatory neuron potentiate or depress, while preserving relative strengths, to maintain a target level of activity [5, 6]. An alternative mechanism that has gained much recent attention is heterosynaptic plasticity [7, 8], which occurs both at excitatory and inhibitory synapses that have not been directly affected by the induction of plasticity [9]. A third plausible homeostatic mechanism with significant experimental evidence is intrinsic plasticity which affects the intrinsic excitability of single neurons by adjusting the distribution of different ion channel subtypes [10, 11].

Various computational studies have benefited from this plethora of experimental evidence for homeostatic control of firing rates and synaptic strengths, and implemented a range of computational models from purely phenomenological ones to detailed biophysical ones. Some relatively straightforward ways to stabilize firing rates and control synaptic strengths in models include imposing upper bounds on synaptic strengths, applying normalization schemes which adjust synaptic strengths by preserving the total sum of incoming weights into a neuron [3, 12] and assuming that the plasticity mechanism modifying synaptic strengths is itself plastic—called 'metaplasticity' [13–16]. These can often be linked to the above experimentally described homeostatic mechanisms. Computational studies have also begun to uncover the various, often complementary, functional roles of different homeostatic mechanisms, e.g. of synaptic scaling versus intrinsic plasticity [16] or heterosynaptic plasticity [9]. However, how exactly synaptic plasticity and homeostatic mechanisms interact to control synaptic strengths, and yet enable learning, is still partially unresolved [17–19]. Part of the challenge is that the experimentally measured timescales of synaptic scaling are too slow to stabilize the Hebbian runaway dynamics in computational models, where much faster normalization schemes are used instead [16, 20–23]. This is sometimes referred to as the 'temporal paradox' of homeostasis [24–26]. A related problem to the integration of plasticity and homeostasis is the trade-off between stability and flexibility. While stimulus representations need to be stable, for instance to allow long-term memory storage, the system also needs to be flexible to allow re-learning of the same, or learning of new representations [27]. This has been successfully achieved in some circumstances. For example, implementing metaplasticity in the excitatory connections through a sliding threshold between potentiation and depression can generate weight selectivity and firing rate stability [13, 14, 16, 24]. Additionally, heterosynaptic plasticity has been modeled to stabilize synaptic weight dynamics, while still allowing learning [9, 28–30], including behavioral learning [31]. A strong candidate for stabilizing synaptic weights is the induction of homosynaptic LTP (LTD) together with heterosynaptic LTD (LTP) at nearby synapses, referred to as the 'Mexican hat' profile of homo- and heterosynaptic plasticity [32, 33].

Here, we investigate an alternative, under-explored mechanism to control and stabilize excitatory synaptic strengths and their dynamics: long-term plasticity of inhibitory-to-excitatory (I-to-E) synapses, also referred to as inhibitory plasticity. Experimental paradigms have characterized diverse forms of inhibitory plasticity, usually via high-frequency stimulation

[34–36] and via pairing of presynaptic and postsynaptic spikes [37, 38]. Inhibition has been shown to control the plasticity mechanisms regulating connection strengths between excitatory neurons depending on their firing rates [39] as well as precise spike timing [40–42]. Inhibitory plasticity can even dictate the direction of excitatory plasticity, shifting between depression or potentiation [43]. Computational models have shown that different forms of inhibitory plasticity can stabilize excitatory rates [44–46]. Given this potential of inhibitory plasticity to affect so many different aspects of synaptic strength and firing rate dynamics in a network, it remains unclear what properties are important for achieving stability, while still enabling neural circuits to learn.

Using computational modeling, we characterize a novel mechanism of inhibitory plasticity with two key features. First, we propose that inhibitory plasticity should depend nonlinearly on the firing rate of an excitatory postsynaptic neuron to robustly control and stabilize the strengths of excitatory synaptic connections made by that neuron. This means that for low postsynaptic rates, I-to-E synapses should depress, for high postsynaptic rates I-to-E synapses should potentiate and without any postsynaptic activity undergo no plasticity. This nonlinear dependence of inhibitory plasticity on the postsynaptic firing rate is sufficient for stability, without the need for additional homeostatic mechanisms. Second, we require a dominance of inhibition, which can be reflected in the larger number of synaptic connections, faster plasticity dynamics of inhibitory synapses or overall higher firing rates of inhibitory neurons relative to excitatory ones. Dominance of inhibition has already been demonstrated in circuits in the visual cortex which operate as inhibition-stabilized networks (ISNs) [47–49]. A direct consequence from our proposed novel mechanism of nonlinear inhibitory plasticity is the emergence of a fixed ratio of excitatory-to-inhibitory synaptic strengths when input rates are constant, in agreement with experimental data [37]. Besides stability, our proposed inhibitory plasticity mechanism can also support flexible learning of receptive fields and recurrent network structures by gating excitatory plasticity via disinhibition [50, 51]. Therefore, our results provide a plausible solution to the stability-flexibility problem by identifying key aspects of inhibitory plasticity, which provide experimentally testable predictions.

## Results

### A linear inhibitory plasticity rule fails to robustly stabilize weight dynamics

To investigate the plausibility of inhibitory plasticity as a control mechanism of excitatory synaptic strengths, we initially considered a model based on a feedforward inhibitory motif prominent in many brain circuits (Fig 1A). Here, a population of presynaptic excitatory neurons projects to a population of inhibitory neurons and both populations project to a single postsynaptic excitatory neuron. Such a motif could resemble, for instance, the excitatory input from the thalamus to excitatory and inhibitory neurons in a primary sensory cortical area [52]. We described the activity of neurons by their firing rates. We considered a network consisting of an excitatory postsynaptic neuron with a linear threshold transfer function and firing rate $v^E$, receiving input from $N^E$ excitatory presynaptic neurons (each with index $j$) with firing rates $\rho_j^E$ through excitatory weights $w_j^{EE}$, and from $N^I$ inhibitory presynaptic neurons (each with index $k$) with firing rates $v_k^I$ through inhibitory weights $w_k^{EI}$:

$$\tau_{FR}^E \dot{v}^E = -v^E + \left[ \sum_{j=1}^{N^E} \rho_j^E w_j^{EE} - \sum_{k=1}^{N^I} v_k^I w_k^{EI} \right]_+ , \tag{1}$$

where $[]_+$ denotes a rectification that sets negative values to zero. The inhibitory neurons follow similar dynamics and are driven by the same $N^E$ presynaptic excitatory neurons with firing

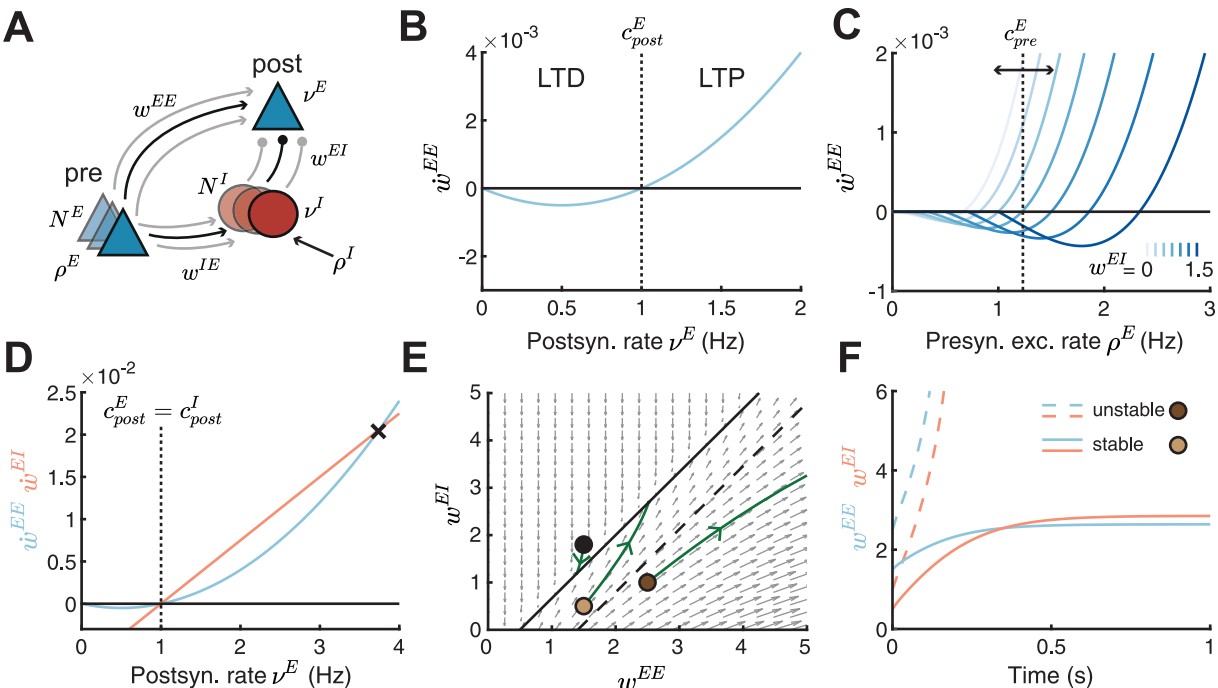

**Fig 1. Linear inhibitory plasticity fails to stabilize weights for high postsynaptic firing rates. A.** Schematic of a feedforward inhibitory motif. A single postsynaptic excitatory neuron with rate $\nu^E$ receives input from $N^E$ excitatory presynaptic neurons, with firing rate $\rho^E$ and weight $w^{EE}$ and $N^I$ inhibitory presynaptic neurons, with firing rate $\nu^I$ and weight $w^{EI}$. The inhibitory neurons receive external excitatory input with rate $\rho^I$ and input from the presynaptic excitatory neurons via $w^{IE}$. **B.** Plasticity curve of E-to-E weights ($\dot{w}^{EE}$, blue) as a function of the postsynaptic rate $\nu^E$. The postsynaptic LTD/LTP threshold $c^E_{post}$ is set to 1. **C.** E-to-E weight change ($\dot{w}^{EE}$) as a function of the presynaptic excitatory rate $\rho^E$ for different I-to-E weights $w^{EI}$ ranging from 0 to 1.5. The presynaptic LTD/LTP threshold $c^E_{pre}$ is shown for $w^{EI} = 0.75$ (vertical dashed line). **D.** Plasticity curves of E-to-E ($\dot{w}^{EE}$, blue) and I-to-E ($\dot{w}^{EI}$, red) weights as a function of the postsynaptic rate $\nu^E$. The excitatory and inhibitory LTD/LTP thresholds are identical ($c^E_{post} = c^I_{post}$). The black cross marks the postsynaptic rate where the plasticity curves cross beyond which the weight dynamics become unstable. **E.** Phase portrait of the dynamics of E-to-E ($w^{EE}$) and I-to-E ($w^{EI}$) weights. Gray arrows indicate the direction of weight evolution over time, points represent three different weight initializations, $[w_0^{EE}, w_0^{EI}] = \{[1.5, 1.8], [1.5, 0.5], [2.5, 1]\}$, and green lines represent the weight evolution for each case. The two colored points represent initial weights in F. Black line indicates the line attractor and the dashed line separates stable from unstable initial conditions (Methods, Eq 20). **F.** E-to-E ($w^{EE}$, blue) and I-to-E ($w^{EI}$, red) weights as a function of time for stable (solid lines, $[w_0^{EE}, w_0^{EI}] = [1.5, 0.5]$) and unstable (dashed lines, $[w_0^{EE}, w_0^{EI}] = [2.5, 1]$) initial conditions.

rates $\rho_j^E$ through excitatory weights $w_j^{IE}$ and additional external input with firing rate $\rho_k^I$,

$$\tau_{FR}^I \dot{\nu}_k^I = -\nu_k^I + \left[ \sum_{j=1}^{N^E} \rho_j^E w_j^{IE} + \rho_k^I \right]_+ . \qquad (2)$$

Here, $\tau_{FR}^E$, $\tau_{FR}^I$ denote the time constants of the firing rate dynamics. For simplicity, we do not use subscripts for neuron identity and interpret all variables as mean values and hence can denote the total excitatory input to the postsynaptic neuron as $N^E \rho^E w^{EE}$ and the total inhibitory input as $N^I \nu^I w^{EI}$. The synaptic weights $w^{EE}$ and $w^{EI}$ are plastic according to different plasticity rules (see below).

Experimental studies have shown that the sign and magnitude of excitatory plasticity depends nonlinearly on the firing rates [53–55]. Inspired by these findings, we implemented plasticity of E-to-E synaptic connections $w^{EE}$ (or weights) as a nonlinear function of the postsynaptic rate $\nu^E$ (Fig 1B):

$$\tau_w^E \dot{w}^{EE} = \rho^E \nu^E (\nu^E - c^E_{post}). \qquad (3)$$

Here, $\rho^E$ denotes the excitatory presynaptic rate and $\tau_w^E$ is the timescale of excitatory plasticity. We refer to the postsynaptic rate at which the plasticity changes sign as the 'postsynaptic LTD/LTP threshold', denoted by $c_{post}^E$. If the firing rate $v^E$ is smaller than the threshold $c_{post}^E$, then the change in synaptic strength is negative leading to long-term depression (LTD), while if $v^E$ is larger than $c_{post}^E$, then the change in synaptic strength is positive leading to long-term potentiation (LTP) (Fig 1B and S1 Fig). This means that increasing the excitatory postsynaptic firing rate will lead to potentiation of excitatory weights, and in a positive feedback loop will further increase the neuron's firing rate—known as the classical problem of 'Hebbian runaway dynamics'.

Hence, we wanted to determine a plausible mechanism to counteract excitatory runaway dynamics. We postulated that regulating the inhibitory input into the postsynaptic neuron provides an efficient way to stabilize excitatory weights and firing rates. In our framework, inhibitory neurons can affect excitatory plasticity in three equivalent ways. (1) The number of inhibitory synapses $N^I$ onto the postsynaptic neuron can change, for example, through the growth or removal of synapses via structural plasticity. (2) The strength of I-to-E synapses $w^{EI}$ can change via inhibitory plasticity. (3) Finally, the rate of inhibitory neurons $v^I$ can also change through the external excitatory input to the inhibitory neurons $\rho^I$ or the excitatory-to-inhibitory weight $w^{IE}$. Various experimental studies have revealed that the plasticity of I-to-E synapses can be induced via the stimulation of the relevant input pathways [34, 35, 43]. Given this experimental evidence for the plasticity of I-to-E synapses, we examined the influence of changing the strength of I-to-E synapses, $w^{EI}$, on the strength and magnitude of E-to-E synapses, $w^{EE}$ (Fig 1C).

We found that stronger $w^{EI}$ weights rates require higher presynaptic excitatory rates to induce LTP, while weaker $w^{EI}$ weights require lower presynaptic excitatory rates to induce LTP. This effectively leads to a shift of the threshold between LTD and LTP as a function of the presynaptic excitatory firing rate as $w^{EI}$ changes. We refer to the presynaptic excitatory firing rate at which the plasticity changes sign between potentiation and depression as the 'presynaptic LTD/LTP threshold', denoted by $c_{pre}^E$ (Fig 1C). In contrast to the fixed postsynaptic LTD/LTP threshold, $c_{post}^E$ (Fig 1B), this presynaptic LTD/LTP threshold depends, among others, on the strength of I-to-E synapses (Fig 1C; Methods, Eq 13).

Rather than hand-tuning the strength of I-to-E synapses, here we propose that a particular inhibitory plasticity rule can dynamically adjust their strength as a function of presynaptic inhibitory and postsynaptic excitatory activity. However, the exact form of this plasticity has not yet been mapped experimentally. Therefore, we first investigated an inhibitory plasticity rule widely-used in computational models which depends linearly on the postsynaptic rate $v^E$ [44, 56] (Fig 1D, $\dot{w}^{EI}$):

$$\tau_w^I \dot{w}^{EI} \quad = v^I(v^E - c_{post}^I). \tag{4}$$

Here, $\tau_w^I$ denotes the timescale of inhibitory plasticity. As for excitatory plasticity, we refer to the postsynaptic rate at which inhibitory plasticity changes from LTD to LTP as the 'inhibitory postsynaptic LTD/LTP threshold', denoted by $c_{post}^I$. This threshold determines the 'target rate' of the postsynaptic neuron [44]. If the excitatory postsynaptic neuron fires at higher rates than $c_{post}^I$, inhibitory LTP leads to a decrease of its firing rate, while if the neuron fires at lower rates than $c_{post}^I$, inhibitory LTD increases its rate. To prevent an unstable scenario where excitatory (Eq 3) and inhibitory plasticity (Eq 4) push the postsynaptic excitatory neuron towards two different firing rates, here we assume that the excitatory and inhibitory thresholds are matched (Fig 1D, $c_{post}^E = c_{post}^I$).

To investigate the effect of this 'linear inhibitory plasticity' mechanism on the temporal evolution of excitatory and inhibitory synaptic weights, $w^{EE}$ and $w^{EI}$, we plotted the flow field in the phase plane $w^{EI}$ vs. $w^{EE}$ (Fig 1E). We found that the interaction of excitatory and inhibitory plasticity generates a line of stable fixed points (i.e. a line attractor) where both synaptic weights do not change any more (Fig 1E, black solid line; see Methods). The initial weights determine whether the weights ultimately converge to the line attractor and stabilize. When the initial E-to-E weights $w^{EE}$ are much larger than the initial I-to-E weights $w^{EI}$ (Fig 1E, below the dashed line), the weights become unstable (Fig 1E and 1F). Equivalently, the weights become unstable when the postsynaptic rate $v^E$ is beyond the crossover point of the excitatory and inhibitory plasticity curves as a function of the postsynaptic excitatory rate (Fig 1D, black cross). For firing rates beyond this crossover point, the E-to-E weights increase faster than the I-to-E weights, leading to runaway dynamics.

In summary, our results suggest that a well-known form of inhibitory plasticity with a linear dependence on the postsynaptic excitatory firing rate can control excitatory weight changes only for a range of initial conditions. There exists a whole range of initial conditions (specifically where the E-to-E are larger than the I-to-E weights) where the postsynaptic excitatory firing rate is sufficiently large and where the weight dynamics explode. This scenario could be problematic if during normal development in the animal, the E-to-E and I-to-E weights are set up in this range, and implies the need for careful tuning to prevent unlimited weight growth.

## A novel nonlinear inhibitory plasticity rule as a robust mechanism to stabilize excitatory weights

To ensure weight stability without fine tuning of the initial E-to-E and I-to-E weights, we proposed a novel inhibitory plasticity rule. The rule depends nonlinearly on the postsynaptic rate $v^E$, similarly to excitatory plasticity (Eq 3, Fig 2A):

$$\tau_w^I \dot{w}^{EI} = v^I v^E \left( v^E - c_{post}^I \right). \tag{5}$$

As before, to prevent a scenario where the two, excitatory and inhibitory, plasticity rules push the postsynaptic excitatory neuron towards two different firing rates, we assume here that the excitatory and inhibitory thresholds are matched $c_{post}^E = c_{post}^I$. However, as we show later, this assumption can be relaxed. Differently from the linear inhibitory plasticity rule (Eq 4), the nonlinear inhibitory plasticity rule ensures that I-to-E synapses do not change in the case where the postsynaptic firing rate is zero (Fig 2B, beyond gray line), as shown in experiments where postsynaptic activity or depolarization is needed to induce inhibitory plasticity [43]. Additionally, the nonlinear rule eliminates the region of initial weight configurations in the phase space where the weights grow out of bound; instead the weights converge to the line attractor (Fig 2B). Indeed, the E-to-E weights, I-to-E weights and the postsynaptic rate reach a stable configuration over time (Fig 2C). We calculated the condition leading to stable weight dynamics (Methods, Eqs 14–17) as a function of the excitatory and inhibitory input rates ($v^I, \rho^E$), the number of synapses ($N^E, N^I$) and the timescale of the plasticity mechanisms ($\tau_w^E, \tau_w^I$):

$$\frac{N^I (v^I)^2}{\tau_w^I} > \frac{N^E (\rho^E)^2}{\tau_w^E}. \tag{6}$$

This condition ensures stable weight dynamics whenever inhibition is more 'dominant' than excitation, either by having more inhibitory synapses ($N^I$), higher inhibitory rate ($v^I$), a faster

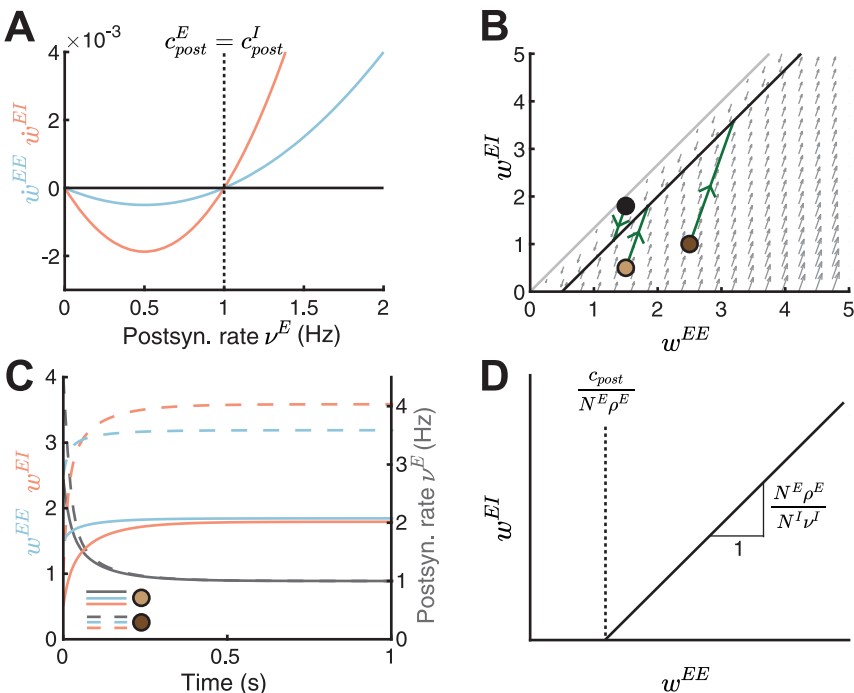

**Fig 2. A novel nonlinear inhibitory plasticity rule can counteract runaway dynamics of excitatory-to-excitatory weights. A.** Plasticity curves of E-to-E ($\dot{w}^{EE}$, blue) and I-to-E ($\dot{w}^{EI}$, red) weights as a function of the postsynaptic rate $\nu^E$. The excitatory and inhibitory LTD/LTP thresholds are identical ($c_{post}^E = c_{post}^I$). **B.** Phase portrait of the dynamics of E-to-E ($w^{EE}$) and I-to-E ($w^{EI}$) weights. Gray arrows indicate the direction of weight evolution over time, points represent three different initial conditions of the weights, $[w_0^{EE}, w_0^{EI}] = \{[1.5, 1.8], [1.5, 0.5], [2.5, 1]\}$, and green lines represent the weight evolution for each initial condition. The two colored points represent initial weights in C. Black line indicates the line attractor and the gray line separates the space at which the postsynaptic firing rate is zero (no dynamics) or larger than zero (Methods, Eq 18). **C.** E-to-E ($w^{EE}$, blue) and I-to-E ($w^{EI}$, red) weight dynamics and postsynaptic rate dynamics ($\nu^E$, gray) as a function of time for two initial conditions in B, $[w_0^{EE}, w_0^{EI}] = [1.5, 0.5]$ (solid lines) and $[w_0^{EE}, w_0^{EI}] = [2.5, 1]$ (dashed lines). **D.** The slope and intersection of the line attractor with the abscissa (black line) depend on the number and firing rates of excitatory and inhibitory neurons and the LTD/LTP threshold.

timescale of inhibitory plasticity ($\tau_w^I$) or a combination thereof. From now on, we assume a faster timescale of inhibitory relative to excitatory plasticity (Methods). An alternative way to achieve stability involves a feedback connection from the postsynaptic neuron to the inhibitory population (S2A Fig). In this case, sufficiently strong E-to-I feedforward and feedback weights guarantee stability in the presence of this feedback inhibitory motif (S2B–S2D Fig).

We found that the line attractor depends on several model parameters (see Methods, Eq 14) (Fig 2D)

$$w^{EI} = \frac{N^E \rho^E}{N^I \nu^I} w^{EE} - \frac{c_{post}}{N^I \nu^I}. \tag{7}$$

Under the assumption that the LTD/LTP thresholds of excitatory and inhibitory plasticity are the same, $c_{post} = c_{post}^E = c_{post}^I$, we found that the slope of the line attractor can be written as $N^E \rho^E/(N^I \nu^I)$, while the intersection of the line attractor with the abscissa can be written as $c_{post}/(N^E \rho^E)$. Therefore, by changing any of the network parameters we can predict the stable configuration to which the weights will converge.

Taken together, we have proposed a novel form of nonlinear inhibitory plasticity which can counteract excitatory runaway weight dynamics without the need for fine tuning. The proposed rule eliminates the need for additional homeostatic mechanisms and upper bounds on

the weights to stabilize weight dynamics. Our modeling approach allows us to dissect the exact dependencies of the stability condition on number of synapses, firing rates and plasticity time-scales of excitatory and inhibitory neurons.

## Dynamic matching of the excitatory and inhibitory postsynaptic thresholds between LTD and LTP

What happens if the postsynaptic thresholds between LTD and LTP for excitatory and inhibitory synapses are not identical, as might be the case in most biological circuits (Fig 3A)? We found that this leads to the disappearance of the line attractor (see Methods Eq 14). When the excitatory postsynaptic threshold is lower than the inhibitory postsynaptic threshold ($c_{post}^E < c_{post}^I$), both E-to-E and I-to-E weighs grow unbounded (Fig 3B). E-to-E weights cannot stabilize as they continue to potentiate ($\dot{w}^{EE} > 0$) even though the postsynaptic neuron is controlled by the fast inhibitory plasticity and approaches the target rate $v^E = c_{post}^I$ (Fig 3C).

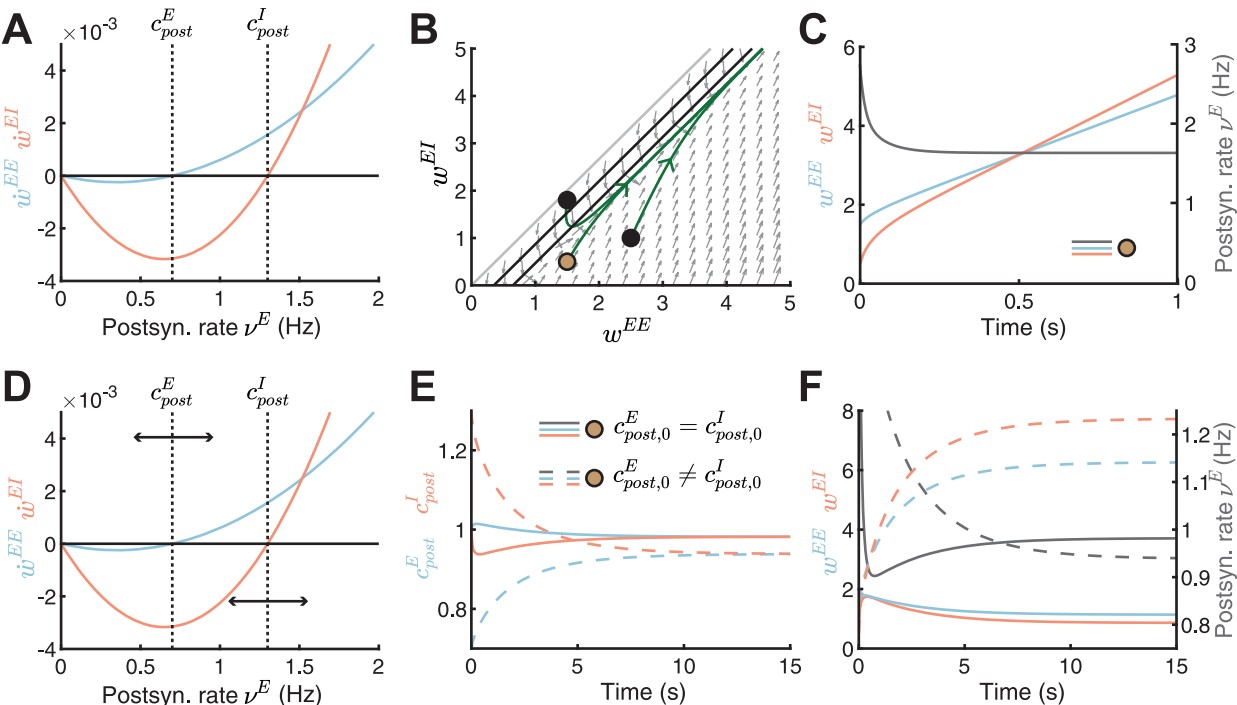

**Fig 3. Dynamic matching of the excitatory and inhibitory postsynaptic LTD/LTP thresholds. A**. Plasticity curves of E-to-E ($\dot{w}^{EE}$, blue) and I-to-E ($\dot{w}^{EI}$, red) weights as a function of the postsynaptic rate $v^E$ with static, non-identical LTD/LTP thresholds ($c_{post}^E = 0.7$, $c_{post}^I = 1.3$). **B**. Phase portrait of the dynamics of E-to-E ($w^{EE}$) and I-to-E ($w^{EI}$) weights for the scenario with static thresholds in A. Gray arrows indicate the direction of weight evolution over time, points represent three different initial conditions of the weights, $[w_0^{EE}, w_0^{EI}] = \{[1.5, 1.8], [1.5, 0.5], [2.5, 1]\}$, and green lines represent the weight evolution for each initial condition. The colored point represents initial weight in C and E-F. Black lines indicate the nullclines and the gray line separates the space at which the postsynaptic firing rate is zero (no dynamics) or larger than zero (Methods, Eq 18). **C**. Excitatory ($w^{EE}$, blue) and inhibitory ($w^{EI}$, red) weight dynamics and postsynaptic rate dynamics ($v^E$, gray) for one initial condition in B, $[w_0^{EE}, w_0^{EI}] = [1.5, 0.5]$. The thresholds are static as in A. **D**. Postsynaptic LTD/LTP thresholds $c_{post}^E$ and $c_{post}^I$ shift dynamically depending on recent postsynaptic rate $v^E$. For lower postsynaptic rate than the excitatory postsynaptic LTD/LTP threshold ($v^E < c_{post}^E$), $c_{post}^E$ decreases, and for $v^E > c_{post}^E$, $c_{post}^E$ increases. For higher postsynaptic rate than the inhibitory postsynaptic LTD/LTP threshold ($v^E > c_{post}^I$), $c_{post}^I$ decreases, and for $v^E < c_{post}^I$, $c_{post}^I$ increases (see Methods). **E**. Evolution of excitatory ($c_{post}^E$, blue) or inhibitory ($c_{post}^I$, red) postsynaptic LTD/LTP thresholds for two different initial conditions ($c_{post,0}^E = c_{post,0}^I$, full lines and $c_{post,0}^E = 0.7$, $c_{post,0}^I = 1.3$, dashed lines). Same initial weight condition as in C, $[w_0^{EE}, w_0^{EI}] = [1.5, 0.5]$, but for dynamic thresholds shown in D. **F**. Excitatory ($w^{EE}$, blue) and inhibitory ($w^{EI}$, red) weight dynamics and postsynaptic rate dynamics ($v^E$, gray) for two different initial conditions ($c_{post,0}^E = c_{post,0}^I$, full lines and $c_{post,0}^E = 0.7$, $c_{post,0}^I = 1.3$, dashed lines). Same initial weight condition as in C, $[w_0^{EE}, w_0^{EI}] = [1.5, 0.5]$, but for dynamic thresholds shown in D. See E for the legend.

Therefore, stability of firing rates does not imply stability of synaptic weights, especially in the case when the postsynaptic thresholds between LTD and LTP are non-equal. In the case of $c_{post}^E > c_{post}^I$, E-to-E and I-to-E weights steadily decrease.

Motivated by experimental findings and theoretical considerations that the excitatory threshold can slide [13, 57], here we proposed that the inhibitory threshold can also be dynamically regulated with both excitatory and inhibitory thresholds shifting into opposite directions (Fig 3D; see Methods). When the postsynaptic rate is lower than the excitatory postsynaptic LTD/LTP threshold ($v^E < c_{post}^E$), the excitatory postsynaptic LTD/LTP threshold should decrease, while when the postsynaptic rate is higher than the threshold ($v^E > c_{post}^E$), the excitatory threshold should increase. Similarly, when the postsynaptic rate is higher than the inhibitory postsynaptic LTD/LTP threshold ($v^E > c_{post}^I$), the inhibitory postsynaptic LTD/LTP threshold should decrease, while when the postsynaptic rate is lower than the threshold ($v^E < c_{post}^I$), the inhibitory threshold should increase. Eventually, these dynamics lead to the matching of excitatory and inhibitory LTD/LTP thresholds (Fig 3E). Therefore, the rates and weights can both be simultaneously stabilized (Fig 3F). The excitatory and inhibitory LTD/LTP thresholds can be matched, and the postsynaptic firing rate and synaptic weights stabilized also for other initializations of the LTD/LTP thresholds (S3A–S3C Fig). Implementing this dynamic threshold adjustment process generates different postsynaptic LTD/LTP threshold configurations (Fig 3E) and postsynaptic rates (Fig 3F, gray lines). Therefore, for different initializations of the LTD/LTP thresholds, a wide variety of stable postsynaptic rates is possible.

## The nonlinear inhibitory plasticity rule can regulate the network response to perturbations

Excitatory and inhibitory LTD/LTP thresholds can be dynamically matched under most conditions, even if they are unequal (S3 Fig). Therefore, from now on we assumed that they are equal and static (as shown in Fig 2A). Next, we wanted to investigate how the new nonlinear inhibitory plasticity rule adjusts the network response following a perturbation. Inspired by sensory deprivation experiments [53, 54, 58] or direct stimulation of input pathways [59, 60], we investigated the network response to perturbing the excitatory presynaptic input rate (Fig 4A).

Independent of the direction of the perturbation, we found that the nonlinear inhibitory plasticity rule brings the excitatory postsynaptic rate back to the target rate (Fig 4B). The inhibitory rate $v^I$ also readjusts because the inhibitory population receives input from the perturbed excitatory population. But the new inhibitory rate is different than the rate before the perturbation (Fig 4B). We found that a perturbation which decreases the excitatory input rate, leads to the depression of both type of weights $w^{EE}$ and $w^{EI}$; in contrast, a perturbation which increases the excitatory input rate leads to their potentiation (Fig 4C). The firing rate response and synaptic weight changes to these perturbations are consistent with previous experimental results [61–66]. Since we used a threshold-linear neuron model (Eqs 1 and 2), our framework can even predict the steady values of the E-to-E and I-to-E synaptic weights, as well as their ratio, by calculating the line attractor in the phase space of $w^{EE}$ and $w^{EI}$ weights as a function of the perturbed parameter (Fig 4D).

Interestingly, we observed that this adjustment occurs by modulation of the presynaptic threshold between LTD and LTP for both excitatory and inhibitory plasticity. Decreasing the excitatory input rate decreases the excitatory presynaptic LTD/LTP threshold, hence limiting the range of presynaptic firing rates that generate depression. The reduction in the LTD/LTP

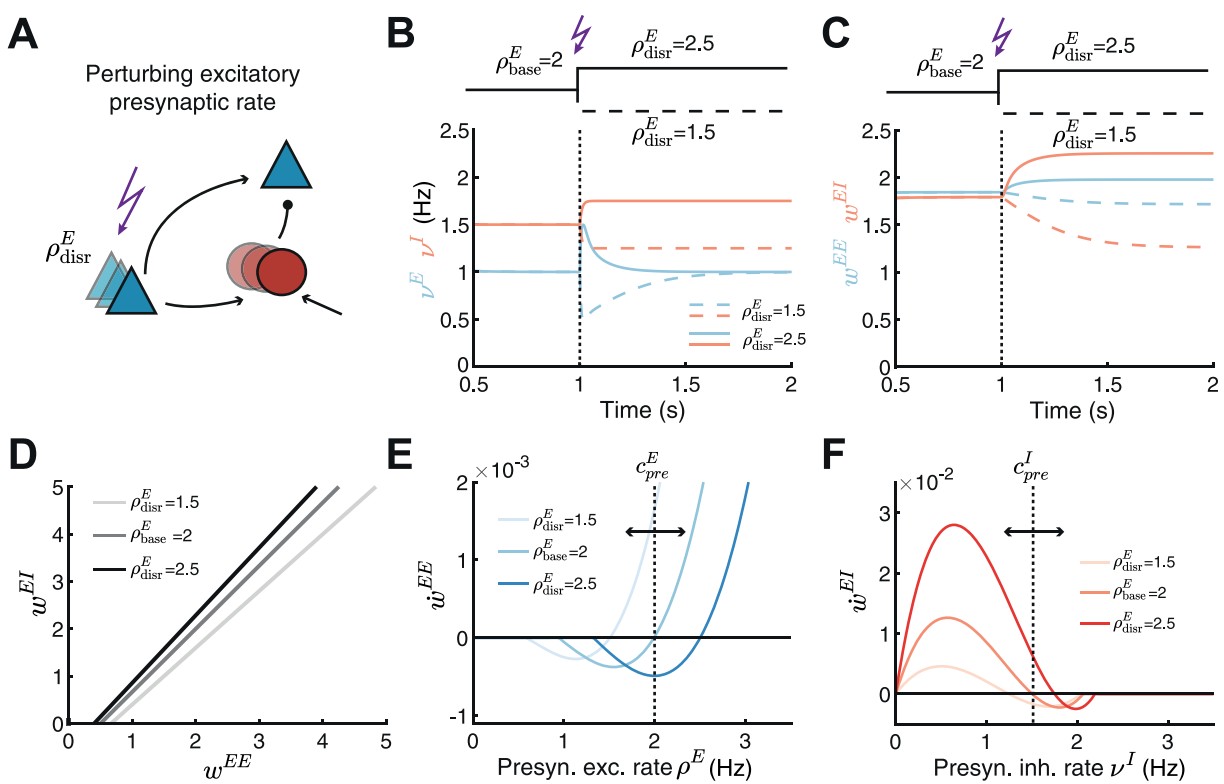

**Fig 4. Nonlinear inhibitory plasticity can regulate the network response to perturbations. A**. Schematic of perturbing the excitatory presynaptic rate in the inhibitory feedforward motif. We use the nonlinear inhibitory plasticity rule with identical excitatory and inhibitory LTD/LTP thresholds from Fig 2A. **B**. Effect of increasing (solid lines, $\rho_{\mathrm{disr}}^E = 2.5$) or decreasing (dashed lines, $\rho_{\mathrm{disr}}^E = 1.5$) excitatory input rates from a baseline of $\rho_{base}^E = 2$ on excitatory (blue) and inhibitory (red) firing rates. **C**. Same as B but for the $w^{EE}$ and $w^{EI}$ weights. **D**. The line attractor for the baseline input $\rho_{\mathrm{base}}^E$ and two input perturbations $\rho_{\mathrm{disr}}^E$. **E**. E-to-E weight change $\dot{w}^{EE}$ as a function of the presynaptic excitatory rate $\rho^E$ for the baseline input $\rho_{\mathrm{base}}^E$ and for two input perturbations $\rho_{\mathrm{disr}}^E$. **F**. I-to-E weight change $\dot{w}^{EI}$ as a function of the inhibitory rate $\nu^I$ for the baseline input $\rho_{\mathrm{base}}^E$ and for two input perturbations $\rho_{\mathrm{disr}}^E$.

threshold follows from the relatively stronger depression of inhibitory compared to excitatory weights allowing the excitatory postsynaptic neuron to fire at the target rate even when the excitatory input is decreased. In contrast, we found that increasing the excitatory input rate increases the LTD/LTP threshold (Fig 4E). Such a shift in the plasticity threshold for excitatory synapses based on presynaptic activity has been measured in sensory deprivation experiments [53, 54, 58], and while restoring vision after sensory deprivation [54, 55] (although deprivation-induced effects occur on much slower timescales than in our plasticity model, see Discussion). Similarly to excitatory plasticity, perturbations in the excitatory input rate also shift the presynaptic threshold between LTD and LTP for inhibitory plasticity (Fig 4F). Since there is no experimental evidence for this effect, we propose it as a prediction for the shift between LTD and LTP for I-to-E weights ($w^{EI}$) in the presence of these perturbations. Even when implementing the plasticity rules with dynamic thresholds, performing the perturbations still leads to stable weight and rate configurations (S3D–S3F Fig).

In summary, the proposed nonlinear inhibitory plasticity can adjust the network response and synaptic strengths to excitatory input rate perturbations, similar to experimental findings. We predict that this shift occurs by modulating the presynaptic LTD/LTP thresholds for both excitatory and inhibitory plasticity.

### The nonlinear inhibitory plasticity rule affects the excitatory-to-inhibitory weight ratio

We next wanted to investigate plausible functional roles of the newly proposed nonlinear inhibitory plasticity besides controlling excitatory and inhibitory firing rates and weights. Given our ability to calculate the steady states of the weights having used a linearly rectified neuron model (Fig 4D), we studied the ratio of E-to-E and I-to-E weights:

$$R^{E/I} = \frac{w^{EE}}{w^{EI}} = \frac{N^I v^I w^{EI} + c_{post}}{N^E \rho^E w^{EI}} = \frac{N^I(N^E \rho^E w^{IE} + \rho^I)w^{EI} + c_{post}}{N^E \rho^E w^{EI}} \tag{8}$$

with $v^I = N^E \rho^E w^{IE} + \rho^I$ (Methods). For strong I-to-E weights $w^{EI}$, the E/I weight ratio approximates to:

$$R^{E/I}_\infty = \frac{N^I v^I}{N^E \rho^E} = \frac{N^I(N^E \rho^E w^{IE} + \rho^I)}{N^E \rho^E} \tag{9}$$

(Fig 5A, inset; see Methods). Therefore, the E/I weight ratio is mainly determined by the ratio of excitatory and inhibitory input rates and the number of synapses, and is independent of the plastic synaptic weights ($w^{EE}$ and $w^{EI}$). A fixed E/I weight ratio can be reached when the input

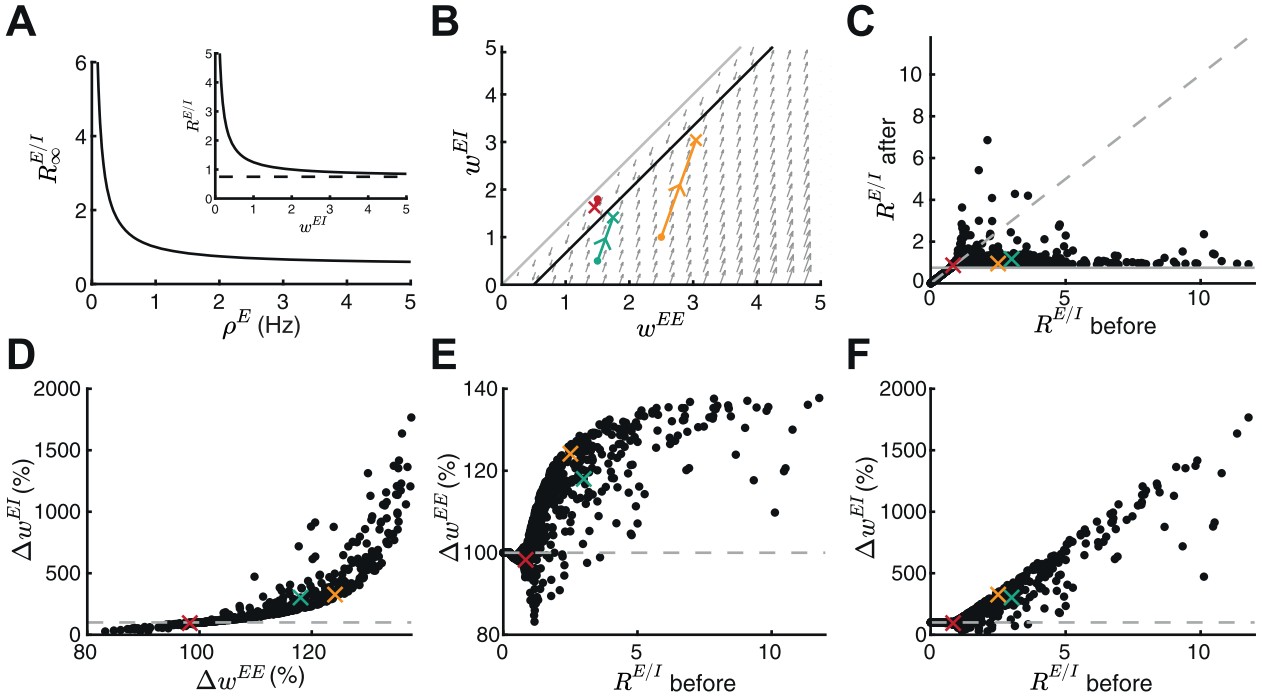

**Fig 5. The nonlinear inhibitory plasticity rule maintains an excitatory-to-inhibitory weight ratio. A.** The steady state E/I weight ratio $R^{EI}_\infty$ as a function of the presynaptic excitatory rate $\rho^E$. Inset: $R^{E/I}$ approaches the steady state $N^I v^I/(N^E \rho^E)$ (dashed line) for large I-to-E weights. **B-F** Based on a random initial weight configuration drawn from a uniform distribution in the range of [0, 3], excitatory and inhibitory plasticity was induced for 100 ms. Extreme initial E/I ratios ($R^{E/I}$ before $> 12$) were excluded from the analysis. **B.** Phase portrait of the dynamics of E-to-E ($w^{EE}$) and I-to-E ($w^{EI}$) weights. Gray arrows indicate the direction of weight evolution over time, colored points represent three different weight initialization, $[w_0^{EE}, w_0^{EI}] = \{[1.5, 1.8], [1.5, 0.5], [2.5, 1]\}$, colored lines represents the weight evolution for each case and the cross marks the weights after plasticity induction. The firing rates dynamics are similar as in Fig 2. **C.** E/I ratio before and after plasticity induction. Crosses indicate examples in B. Gray dashed line indicates the identity line and gray line indicates $R^{E/I}_\infty$. **D.** E-to-E weight change $\Delta w^{EE}$ versus I-to-E weight change $\Delta w^{EI}$ after plasticity induction in percent of initial synaptic weights. Dashed gray line indicates initial I-to-E weight strength and crosses indicate examples in B. **E.** E-to-E weight change $\Delta w^{EE}$ as a function of E/I ratio $R^{E/I}$ before plasticity in percent of initial weights. Dashed gray line indicates initial E-to-E weight strength and crosses indicate examples in B. **F.** Same as E but for I-to-E weight change $\Delta w^{EI}$.

rates are constant. The E/I ratio decreases as the presynaptic excitatory rate $\rho^E$ increases (Fig 5A; Eq 8). This can be explained by considering that a higher excitatory input rate $\rho^E$ generates more excitatory LTP (Fig 1C), which is counteracted by even more inhibitory LTP to stabilize weight dynamics. Analytically, this corresponds to a line attractor with a steeper slope (Figs 2D and 4D for increasing $\rho^E$) since the E/I ratio $R_\infty^{E/I}$ corresponds to the slope of the line attractor (Fig 2D; Methods).

Inspired by experiments [37], we evaluated the E/I ratio $R^{E/I}$ before and after inducing excitatory and inhibitory plasticity for multiple initial weight configurations (Fig 5B and 5C; Methods). As predicted analytically (Fig 5A), the E/I ratio after plasticity in these simulations approaches $R_\infty^{E/I}$ (Fig 5C), matching experiments in the mouse auditory cortex where inducing excitatory and inhibitory plasticity generates a fixed E/I ratio [37]. Large E/I ratios before plasticity induction show the most drastic changes, with high postsynaptic firing rates resulting from dominant excitation needing to be overcome by fast and drastic weight changes by nonlinear inhibitory plasticity. Indeed, we observed that the I-to-E weights exhibit more change than E-to-E weights (Fig 5D). This suggests that nonlinear inhibitory plasticity affects the E/I ratio more prominently than excitatory plasticity (Fig 5E and 5F). With the linear inhibitory plasticity rule [44], a fixed E/I ratio for constant input rates is only reached for initial weights which ultimately converge to the line attractor (Fig 1E).

## Performance of the nonlinear inhibitory plasticity rule under varying presynaptic input and postsynaptic firing rate

We next investigated the effect of varying the presynaptic input or the postsynaptic firing rate on the stability of weight dynamics. Adding noise or a sinusoidal input to the postsynaptic firing rate $\nu^E$ (Methods) maintains synaptic weights within a certain range despite fluctuations (Fig 6A and 6B). We can understand the weight dynamics by studying how a varying input to the postsynaptic neuron affects the line attractors in the phase plane of the $w^{EE}$ and $w^{IE}$ weights. Adding an input to the postsynaptic neuron shifts only the point where the line attractor intersects the abscissa but does not change the slope (Fig 6C; Methods). Therefore, the weights remain constrained within a narrow region, without runaway dynamics. Even when implementing the plasticity rules with dynamic thresholds, adding postsynaptic noise or sinusoidal input leads to stable weight and rate configurations (S4 Fig).

The picture changes when the presynaptic input rate $\rho^E$ varies (Methods). Here, both excitatory and inhibitory weights begin to slowly drift towards higher values while average firing rates remain stable (Fig 6D and 6E). The drift is due to a change in the presynaptic rate which affects the slope of the line attractors (see also Figs 2D and 4D). In the case of presynaptic sinusoidal input rate, the weights slowly increase while oscillating between the line attractors (Fig 6F). Therefore, while on a short timescales the interaction of only excitatory and inhibitory plasticity mechanisms seems to be sufficient to regulate weight and rate stability, we suggest that additional homeostatic mechanisms are necessary to regulate synaptic weight dynamics over longer timescales in the presence of noise or variability in the presynaptic input.

## Gating of receptive field formation via a disinhibitory signal

What functional implications does the proposed nonlinear inhibitory plasticity rule have on setting up network circuitry? Other than controlling excitatory and inhibitory rates and weights, here we wanted to examine if the nonlinear inhibitory plasticity rule can also enable flexible learning. Various forms of synaptic plasticity have been observed to support receptive field formation and generate selectivity to stimulus features in the developing cortex [67]. To investigate the function of interacting excitatory and inhibitory plasticity at the network level,

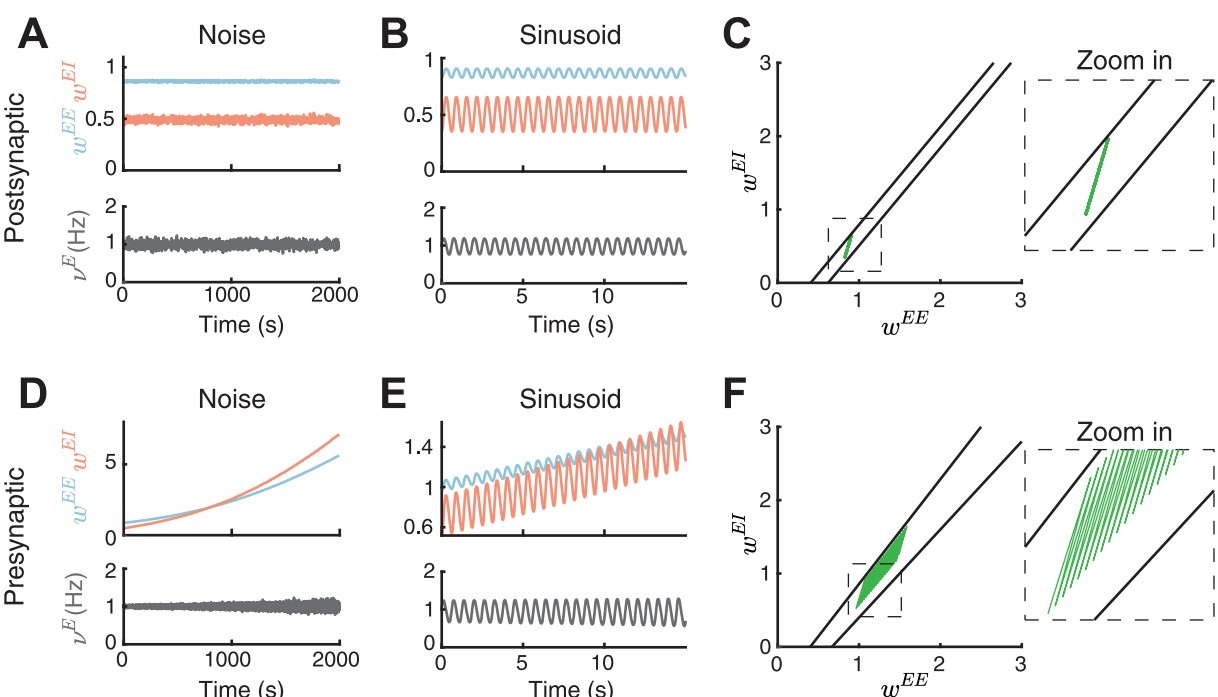

**Fig 6. Performance of the nonlinear inhibitory plasticity rule under varying presynaptic input and postsynaptic firing rate. A**. Adding noise to the postsynaptic firing rate. Top: E-to-E ($w^{EE}$, blue) and I-to-E ($w^{EI}$, red) as a function of time. Bottom: Postsynaptic rate dynamics ($v^E$, gray) as a function of time. **B**. Same as A but after adding a sinusoidal input to the postsynaptic firing rate. **C**. Left: The line attractors in the $w^{EE}$ and $w^{EI}$ phase plane at the maximum and minimum of the postsynaptic firing rate after the addition of sinusoidal input (black lines) and the weight dynamics from B (green). Right: Zoom in of the phase plane. **D**. Same as A but after adding the noise to the presynaptic input rate. **E**. Same as B but after using a sinusoid for the presynaptic input rate. **F**. Same as C but after using a sinusoid for the presynaptic input rate with weight dynamics from E (green).

we first extended the feedforward circuit motif to two independent pathways with pathway-specific inhibition (Fig 7A). We found that perturbing the presynaptic excitatory rate of both inputs in opposite directions, decreasing for input 1 and increasing for input 2, differently shifts the input-specific excitatory presynaptic LTD/LTP thresholds and establishes different E/I ratios (Fig 7B). This shift in the model is in agreement with experimental studies in the hippocampus which have shown that the thresholds between the induction of LTD and LTP are synapse-specific [59, 68]. These results suggest that the control of E-to-E weight dynamics via nonlinear inhibitory plasticity is input-specific.

Applying disinhibition by inhibiting the inhibitory population is a widely considered mechanism to 'gate' learning and plasticity [50, 51, 69]. To test the potential of the circuit with non-linear inhibitory plasticity to learn, we applied a disinhibitory signal by decreasing the external excitatory input onto the inhibitory populations. We found that this decreases the inhibitory input onto the postsynaptic neuron and potentiates E-to-E synapses, $w^{EE}$ (Fig 7C, $\rho^I < 1$). In contrast, increasing the input onto the inhibitory populations depresses E-to-E synapses (Fig 7C, $\rho^I > 1$). Therefore, disinhibition via perturbation of the inhibitory neurons has the capacity to induce plasticity at E-to-E synapses and can gate excitatory plasticity.

How do the current results generalize to larger circuits with multiple independent inputs? In addition to pathway-specific inhibition, in this extended circuit we also introduced an unspecific inhibitory population (Fig 7D). We presented different inputs to each of ten pathways in random order, corresponding to oriented bars in the visual cortex, or different single tone frequencies in the auditory cortex (Methods). We found that disinhibiting via the

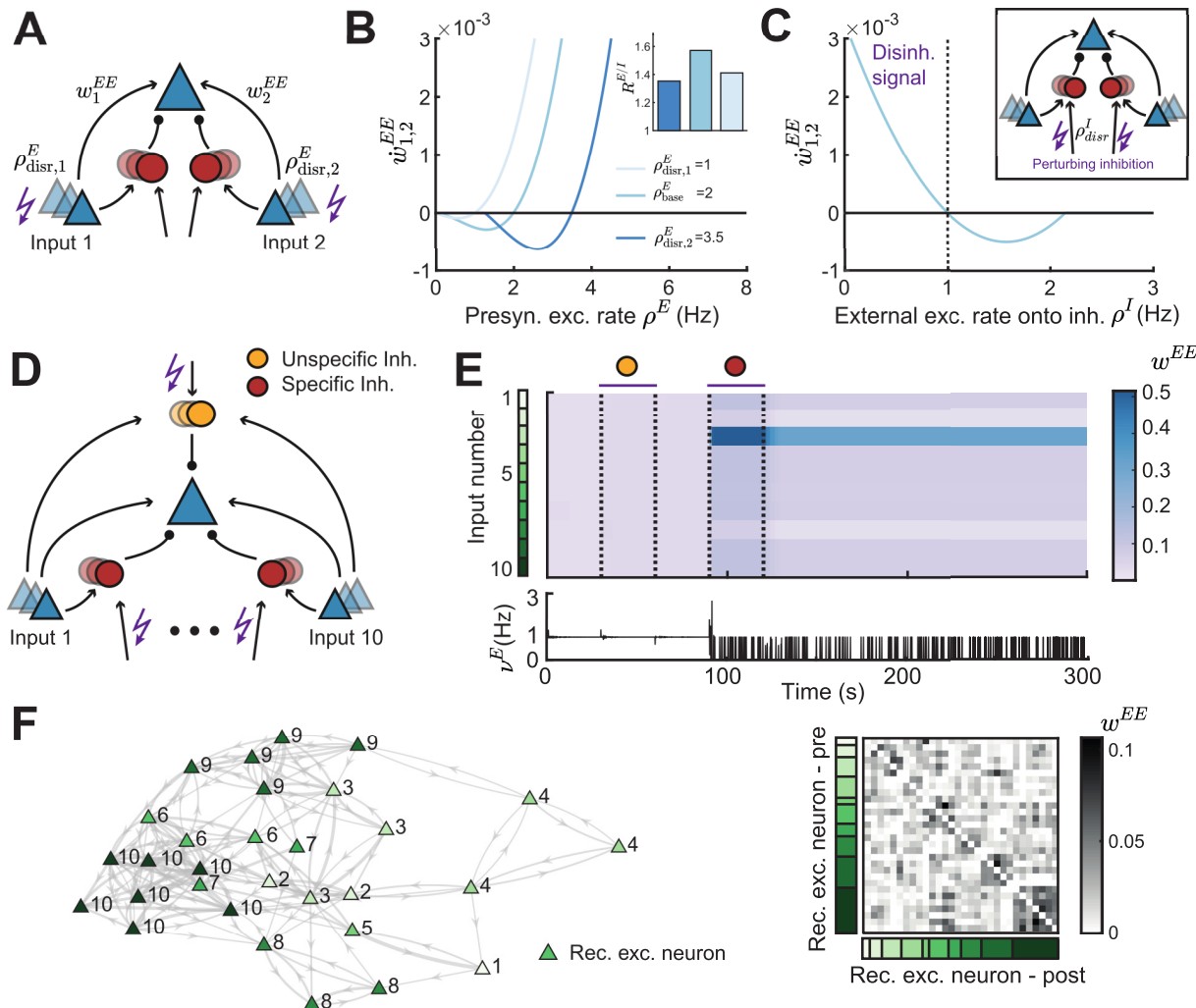

**Fig 7. Gating of receptive field formation via a disinhibitory signal. A**. Two independent inputs onto the same postsynaptic excitatory neuron. We perturb the presynaptic excitatory rate from input 1 or 2 ($\rho^E_{\text{disr},1,2}$). **B**. Plasticity curve of E-to-E weights for input 1 or 2 ($\dot{w}^{EE}_{1,2}$) as a function of the presynaptic excitatory rate $\rho^E$ for different input-specific perturbations $\rho^E_{\text{disr},1,2}$. Inset: E/I weight ratio $R^{E/I}$ for different input-specific perturbations. **C**. Plasticity curve of E-to-E weights for input 1 and 2 ($\dot{w}^{EE}_{1,2}$) as a function of the external excitatory rate onto the inhibitory neurons $\rho^I$, corresponding to a perturbation $\rho^I_{\text{disr}}$ of the inhibitory populations. Perturbing $\rho^I$ below 1 Hz (dashed line) is interpreted as a disinhibitory signal. Inset: We perturb the external excitatory rate onto the inhibitory neurons $\rho^I_{\text{disr}}$. **D**. Ten independent inputs onto the same postsynaptic excitatory neuron with one inhibitory population unspecific to the input (yellow) and ten inhibitory populations each specific to one input (red). **E**. Top: Evolution of excitatory weights over time. Purple bars indicate the time window where either the unspecific (yellow) or all specific (red) inhibitory populations is disinhibited by applying a negative input onto the inhibitory neurons (Methods). Input number color coded in green. Bottom: Postsynaptic firing rate $\nu^E$ over time. **F**. Left: Network connectivity of recurrently connected excitatory neurons (triangles) after disinhibition. The number and the color indicates the input to which each neuron formed a receptive field (10 inputs in total). The thickness of the connection indicates the strength, only weights above 0.03 are shown. Distance and position of neurons is for visualization purposes only. Right: Ordered recurrent E-to-E connectivity matrix. Input number color coded in green as in panel E.

unspecific inhibitory population does not selectively potentiate E-to-E weights, and hence does not generate competition among the different inputs. In this case, the selective potentiation of E-to-E weights corresponding to the inputs stimulated at a given time is counteracted by the potentiation of I-to-E weights specific to the stimulated inputs. This fast cancellation of any input-specific excitatory plasticity by input-specific inhibitory plasticity generates very small changes in the postsynaptic firing rate (Fig 7E, bottom). In contrast, equally disinhibiting

via all ten specific inhibitory populations strongly increases the E-to-E weights corresponding to only a subset of inputs, a process also called receptive field formation (Fig 7E). In this case, the selective potentiation of E-to-E weights corresponding to the inputs stimulated at a given time is counteracted by the potentiation of all unspecific I-to-E weights. Therefore, inhibitory plasticity does not cancel input-specific excitation. The random presentation order of the different inputs generates input-specific differences in excitatory weights and hence leads to competition. The input-specific potentiation is reflected in the fluctuating postsynaptic firing rate which increases only when the winning input is presented (Fig 7E, bottom).

Finally, we implemented a network of 30 recurrently connected excitatory neurons where each neuron in the circuit receives inputs from ten inputs and an unspecific and a specific inhibitory population (as in Fig 7D). In addition to the feedforward excitatory and inhibitory synapses, all recurrent E-to-E weights are also plastic. Similar as with a single postsynaptic neuron, we found that each of the excitatory neurons in the recurrent circuit forms a receptive field by becoming selective to one of the inputs (Fig 7F, left; number next to the neuron). In addition, strong bidirectional connections form among recurrent excitatory neurons with similar receptive fields due to their correlated activity (Fig 7F). This is consistent with strong bidirectional connectivity described in multiple experimental studies [70–72].

In summary, the newly proposed nonlinear inhibitory plasticity rule does not only ensure stable synaptic weights and activity, but also enables the formation of feedforward and recurrent structures upon disinhibition which gates synaptic plasticity.

## Discussion

Hebbian excitatory synaptic plasticity is inherently unstable, requiring additional homeostatic mechanisms to control and stabilize excitatory-to-excitatory weight dynamics [4]. Here, we proposed a novel form of inhibitory plasticity (Fig 2), which can control excitatory and inhibitory firing rates and synaptic weights and enable stable and flexible learning of receptive fields in circuit models of the sensory cortex. We identified the dominance of inhibition over excitation (Eq 6) and identical postsynaptic thresholds between LTD and LTP for excitatory and inhibitory plasticity (compare Fig 2A and Fig 3A–3C) as two necessary features for stabilization of weight dynamics in our model. However, the latter requirement can be relaxed with a suitable dynamic mechanism that enables self-adjusting of the plasticity thresholds in opposite directions for excitatory and inhibitory plasticity (Fig 3D–3F). This novel form of nonlinear inhibitory plasticity can also regulate the network response to perturbations of excitatory input rates (Fig 4). Inhibitory plasticity affects the E/I weight ratio and establishes a fixed E/I ratio when input rates are constant (Eq 8), in agreement with experiments in the mouse auditory cortex where inducing excitatory and inhibitory plasticity sets a fixed E/I ratio [37] (Fig 5). We find that varying the presynaptic inputs or the postsynaptic firing rate differently affects stability (Fig 6). Besides stability, the proposed form of inhibitory plasticity enables receptive field formation following disinhibition to input-specific inhibitory populations and in recurrent networks supports the formation of strong bidirectional connectivity among neurons with similar receptive fields (Fig 7), suggesting a possible solution for the stability-flexibility problem.

### Inhibitory plasticity as a control mechanism of excitatory-to-excitatory weight dynamics

In the last decades, experimental studies have uncovered multiple possible mechanisms to counteract Hebbian runaway dynamics, including synaptic scaling [5, 73], heterosynaptic plasticity [7, 8], and intrinsic plasticity [10, 11]. At the same time, computational studies have

included multiple homeostatic mechanisms, some of them the same as the experimental ones, to stabilize rates and weight dynamics, including upper bounds on the E-to-E weights, normalization mechanisms [3, 12, 16, 20, 21, 23], metaplastic changes of the plasticity function [13–16, 24], heterosynaptic plasticity [9, 29, 30] and intrinsic plasticity and synaptic scaling [16]. However, the spatial and temporal scales for integrating Hebbian and homeostatic plasticity continue to be subject of investigation [18, 25, 26]. This is especially the case for synaptic scaling which experimentally operates on timescales too slow to counteract the faster Hebbian synaptic plasticity (hours and days, vs. seconds and minutes). Heterosynaptic plasticity has been suggested as a more natural solution to the 'temporal paradox' problem since it operates on a similar timescale as Hebbian plasticity [9, 28, 29].

In our study, we instead proposed a novel inhibitory plasticity rule at inhibitory-to-excitatory synapses which depends nonlinearly on the postsynaptic firing rate as a solution to the temporal paradox problem. While nonlinear excitatory plasticity rules have been identified in experimental studies [53–55], less data is available for inhibitory plasticity. For example, presynaptic stimulation (hyperpolarization) and postsynaptic depolarization, have been shown to be required for inhibitory plasticity induction [74–77]. Additionally, high-frequency stimulation of presynaptic input pathways has been shown to potentiate inhibitory synapses [34–36]. Finally, the amount of inhibitory LTP has been shown to depend on the postsynaptic rate [43]. We designed our nonlinear inhibitory plasticity mechanism to be consistent with these findings: both, pre- and postsynaptic activity is necessary to induce inhibitory plasticity and the amount of LTP depends on the postsynaptic rate. Nonetheless, our rule is inconsistent with some experimental data which found no inhibitory plasticity for very high postsynaptic rates [43].

Several computational models have explored the functional roles of inhibitory spike-timing-dependent plasticity (iSTDP) operating at inhibitory-to-excitatory synapses. A commonly investigated plasticity rule has a symmetric learning window, where pre- and postsynaptic spikes close in time lead to LTP, and spikes further apart lead to LTD [44]. Similar symmetric learning windows have been identified experimentally in the auditory cortex [37], in the orbitofrontal cortex [78], and in the hippocampus [77]. Asymmetric learning windows, in which pre-post spike pairs lead to LTP and post-pre spike pairs lead to LTD have been observed in the entorhinal cortex [79], and also used in computational studies [45, 46]. For an inhibitory plasticity rule to successfully stabilize postsynaptic excitatory firing rates, it needs to implement a negative feedback mechanism whereby for high postsynaptic firing rates the inhibitory synaptic strength increases, while for low rates the inhibitory strength decreases, as is the case for our rule as well as others [44–46]. The nonlinear inhibitory plasticity we propose in our study is probably closest to a recent implementation of inhibitory plasticity via the voltage rule [80], since the voltage rule has a nonlinear dependency on postsynaptic firing rates [81].

## Inhibitory plasticity as a metaplastic mechanism

The ability of the proposed nonlinear inhibitory plasticity to control the sign and magnitude of excitatory plasticity resembles metaplasticity, i.e. a plasticity mechanism that is plastic itself [13, 15]. We found that input perturbations modulate the excitatory presynaptic LTD/LTP threshold via a change of the I-to-E weights and inhibitory rates consistent with metaplasticity (Fig 4). Previous computational work has already suggested that a linear inhibitory plasticity rule can implement a metaplastic mechanism [56]. What mechanism underlies the sliding LTD/LTP threshold during the induction of plasticity is still an open question. Some experimental studies have suggested that inhibition can control the sign and magnitude of excitatory plasticity [40, 41, 43, 82]. Most intriguingly, it has been shown that application of

gamma-Aminobutyric acid (GABA) can increase the excitatory LTD/LTP threshold, while blocking GABA can decrease the excitatory LTD/LTP threshold [39], supporting our findings (Fig 1C).

The metaplasticity of excitatory plasticity was first suggested theoretically with the Bienenstock-Cooper-Munro (BCM) rule [13], and was later confirmed in sensory deprivation and restoration experiments [53, 54, 55, 58]. In the BCM rule, the metaplastic mechanism is implemented by a sliding LTD/LTP threshold depending on the excitatory postsynaptic rate [83, 84]. Higher (lower) postsynaptic rates lead to a higher (lower) postsynaptic LTD/LTP threshold making LTP (LTD) induction harder. Various implementations of the BCM rule have demonstrated its ability to achieve weight selectivity and firing rate stability without any inhibitory plasticity [13, 14, 24, 85]. Differently from the BCM model, in our nonlinear inhibitory plasticity rule the metaplastic sliding of the LTD/LTP threshold $c_{pre}^E$ depends on the presynaptic excitatory rate (Fig 1C), whereas the postsynaptic LTD/LTP threshold $c_{post}^E$ is fixed (except in Fig 3D–3F and S3 Fig). This apparent difference can be resolved by assuming that homeostatic mechanisms operate at two different timescales: fast and slow. Slow homeostasis has been linked to synaptic scaling which we (and others, e.g. [57]) hypothesize to be a possible mechanism behind changes in the postsynaptic threshold. It is usually observed on the timescales of many hours to days [6, 86, 87] but can also occur on the timescale of a few hours [88]. Fast homeostasis might be linked to disinhibition and inhibitory plasticity [89], which is induced on the timescale of minutes [9, 37, 90]. We suggest this is the case during sliding of the presynaptic LTD/LTP threshold mediated by our inhibitory plasticity rule. Nonetheless, it is plausible that both, presynaptic and postsynaptic metaplasticity exist in neuronal circuits. An advantage of achieving homeostasis via inhibitory plasticity, rather than a direct influence on the E-to-E weights, might be that there is no interference with stored information in E-to-E connections.

We used the metaplasticity of the nonlinear inhibitory plasticity rule to describe firing rate and weight changes in the model following perturbations of excitatory input (Fig 4) such as during sensory deprivation experiments [53, 54, 58]. For example, the decrease in inhibitory firing rates and weights after decreasing excitatory input in our model is consistent with the decrease in inhibitory activity following sensory deprivation [69, 87, 91]. Specifically, sensory deprivation has been shown to depress inhibitory synaptic strengths, decrease in the number of inhibitory synapses [62–66] (but see [92, 93]) and depress excitatory synaptic strengths [61, 94]. Increasing excitatory input in our model potentiates inhibitory weights, in agreement with experiments where up-regulating activity potentiates I-to-E synapses [95, 96]. We note that the plasticity induced by these sensory deprivation experiments occurs on much longer timescales of hours to days (see e.g. [57, 89]) compared to the shorter plasticity timescales of seconds or minutes in our model, suggesting that other mechanisms than the proposed nonlinear inhibitory plasticity drive the experimentally observed changes. Moreover, in our model we instantaneously and permanently change the input firing rate in contrast to the more complex changes in input patterns occurring during sensory deprivation. Therefore, the applied perturbation in our model could be better related to direct simulation of input pathways when similarly fast LTD/LTP threshold changes have been measured experimentally [59, 60, 97]).

## Key features of the nonlinear inhibitory plasticity rule

For the novel inhibitory plasticity rule to stabilize E-to-E weight dynamics, two key features need to be fulfilled. First, I-to-E weight changes need to be more 'dominant' than E-to-E weight changes (Fig 2). More dominant means that I-to-E weights need to change with a higher magnitude at each time step compared to E-to-E weights, for all postsynaptic rates. If

excitatory plasticity exceeds inhibitory plasticity for a certain postsynaptic rate as in the case of linear inhibitory plasticity, weight dynamics will be unstable (Fig 1D–1F). In our model, dominance of nonlinear inhibitory plasticity is guaranteed by the condition in Eq 6, which involves relative number of synapses, presynaptic rates and plasticity timescales of excitation and inhibition to determine stability. Previous experimental work has reported that inhibitory synapses change more drastically than excitatory synapses [37], but inhibitory plasticity may be delayed relative to excitatory plasticity [50].

Second, the excitatory and inhibitory postsynaptic LTD/LTP thresholds need to be matched for stable weight dynamics, whereby excitatory and inhibitory synaptic change occur in the same direction for a given firing rate (Fig 2A–2C versus Fig 3A–3C). However, implementing a mechanism that dynamically shifts these thresholds in the opposite directions for excitatory vs. inhibitory plasticity based on experimental evidence [57], suggests that this match is not needed at all times. An interesting consequence from this dynamic threshold shift is the ability to achieve a range of firing rates. A limitation of the suggested dynamic threshold matching mechanism is that it is non-local whereby the thresholds for all input pathways converge to the same value. While this can still achieve stable weight dynamics and postsynaptic firing rates (S3G–S3I Fig; Methods), it can no longer induce competition among different inputs. Future work needs to investigate whether a different dynamic matching of excitatory and inhibitory LTD/LTP thresholds, perhaps one that is input-specific, can achieve the stable formation of receptive fields.

We found that the newly proposed nonlinear inhibitory plasticity rule achieves a fixed E/I ratio for constant input rates (Fig 5) in agreement with experimental data in the mouse auditory cortex where the induction of excitatory and inhibitory plasticity established a fixed E/I ratio [37]. We observed that inhibitory plasticity is the more dominant mechanism to achieve this. The dominance of inhibitory plasticity suggests a possible solution to the temporal paradox problem of integrating Hebbian excitatory plasticity and homeostasis [25], eliminating the requirement for additional fast stabilizing mechanisms in our model. While the relative timescales of excitatory and inhibitory plasticity mechanisms remain an open question, most computational models agree on the need for faster inhibitory than excitatory plasticity dynamics [25, 98].

Our framework is robust when noise or a varying input is added to the postsynaptic firing rate but not when the presynaptic rate varies (Fig 6). This suggests that additional homeostatic mechanisms are necessary to robustly counteract drift of synaptic weights when the input or the firing rates vary.

## Functional implications of the nonlinear inhibitory plasticity rule

The interaction of the nonlinear inhibitory and excitatory plasticity in our model and the overlap of excitatory and inhibitory LTD/LTP thresholds lead to a fixed E/I weight ratio when input rates are constant (Fig 5A and 5C and Eq 8). This is consistent with several experimental studies which have suggested that inhibitory plasticity maintains a stable E/I ratio [9, 37, 43, 50, 96, 99–102]. For example, as our model would predict, some studies have found that the amount of inhibitory plasticity depends on the E/I ratio before plasticity induction (Fig 5F) [37, 103]. In these experiments, a change in E/I ratio is observed on the timescale of induction of plasticity (5–10 min) [37]. When we perturb the excitatory input rate as a model of sensory deprivation the E/I ratio increases (Fig 5A), consistent with sensory deprivation experiments [66, 69, 91, 94]. Despite the ability of the new nonlinear inhibitory plasticity rule to establish and maintain E/I balance, we acknowledge that there are various additional mechanisms that contribute, including heterosynaptic plasticity [9].

The emergence of fixed E/I ratio for constant input rates following from the stabilization of postsynaptic rates driven by the novel inhibitory plasticity rule ensures E/I balance. E/I balance is more broadly defined as the proportionality of total excitatory and inhibitory input onto a neuron [104]. In our model, once the neuron fires with a firing rate equal to the LTD/LTP threshold there is no more synaptic plasticity. To induce further weight changes, an additional gating signal is necessary that perturbs the postsynaptic firing rate. In our model, there are three ways to gate plasticity: (1) directly changing the postsynaptic rate (Fig 1B); (2) perturbing the excitatory input pathway (Fig 4); and (3) perturbing the inhibitory population (Fig 7C). The idea that inhibition gates excitatory plasticity is well-documented in the experimental literature [105–107].

Experimentally, both neuromodulation [50, 108] and disinhibitory circuits [51, 90, 109–111] can directly control the activity of inhibitory neurons and lead to excitatory plasticity. Based on this, we investigated the gating of plasticity via a disinhibitory signal in the context of receptive field formation. While receptive field formation has already been demonstrated in multiple computational studies [13, 45, 56], we propose that it can occur solely from the interaction of excitatory and inhibitory plasticity without any additional mechanism to induce competition among different inputs (Fig 7D and 7E). Recurrently connecting multiple postsynaptic excitatory neurons and allowing the connections between them to be plastic leads to receptive field formation of each excitatory neuron in the recurrent circuit and the formation of strong bidirectional connectivity between neurons with similar receptive fields (Fig 7F). This is in agreement with various experimental data indicating that similarly responsive neurons are more strongly connected [70–72, 112]. The formation of strongly recurrently connected neurons, often referred to as assemblies, via synaptic plasticity has been shown in previous computational studies [20–23, 113]. In contrast to our framework, these studies rely on a fast normalization mechanism in addition to excitatory and inhibitory plasticity to reliably learn assemblies.

We found that gating of receptive field formation via disinhibition depends on the specificity of the targeted inhibitory population to the inputs. While disinhibiting the unspecific population does not form receptive fields, disinhibiting all specific inhibitory populations induces competition between different inputs and forms receptive fields. If inhibitory plasticity counteracts excitatory plasticity in an input-specific way, no competition between input pathways can emerge because small biases in the E-to-E weights in one input are immediately balanced by I-to-E weights in the same input. Therefore, disrupting the specific inhibitory populations allows the strengthening of a subset of inputs. This result is similar to the findings of [56], where receptive field formation was shown to depend on the specificity of the inhibitory neurons.

The inhibitory populations in our model can be linked to the two main inhibitory neuron types in the cortex, somatostatin-expressing (SOM) and parvalbumin-expressing (PV) inhibitory interneurons. Specificity of the inhibitory neuron type to excitatory inputs can be interpreted as tuning of the inhibitory neurons to input features. In the visual [114, 115] and the auditory cortex [116], tuning of SOM interneurons is sharper than PV interneurons, although conflicting evidence exists [117]. Therefore, in our model the specific inhibitory neuron type could represent SOM interneurons while the unspecific inhibitory population could represent PV interneurons. Supporting this interpretation of SOM interneurons being the specific inhibitory population, experimental studies find that a suppression of SOM neurons gates excitatory plasticity [106, 111, 118]. In contrast to this interpretation, the specific inhibitory neurons in our model might be interpreted as PV neurons. This is supported by experimental evidence which shows that PV neurons strongly inhibit pyramidal neurons which have similar selectivity [119].

## Predictions

Using rate-based units in our model enabled us to treat it analytically and offered an in-depth mechanistic understanding of the involved processes leading to experimentally testable predictions and making our model assumptions falsifiable. A main feature of our model is that inhibitory plasticity depends nonlinearly on the rate of the postsynaptic excitatory neuron. This can be tested experimentally by inducing inhibitory plasticity while varying the rate of an excitatory neuron and keeping the inhibitory input to this neuron constant. A second feature of our model is that excitatory and inhibitory plasticity have an identical postsynaptic LTD/LTP threshold. This could be tested by inducing plasticity at excitatory and inhibitory pathways onto the same excitatory neuron, and measuring the LTD/LTP thresholds as a function of the rate of that neuron.

Based on the perturbation experiment (Fig 4), we can formulate multiple predictions. First, we hypothesize that the mechanism behind the metaplastic mechanism is a change in the level of inhibition (see Figs 1C and 4E). Therefore, blocking inhibitory plasticity experimentally should also disrupt the metaplastic mechanism. Second, we predict that the shape of inhibitory plasticity as a function of the inhibitory rate is reversed compared to excitatory plasticity, and perturbations of the excitatory input lead to specific metaplastic changes of inhibitory plasticity. Decreasing the excitatory input should lower the inhibitory LTD/LTP threshold as a function of the presynaptic inhibitory rate and decrease the inhibitory LTP magnitude (Fig 4F). Third, since the line of stable fixed point depends on several model parameters (Fig 2C and Eq 7), especially on the excitatory input rate (Fig 4D), we hypothesize that different E/I ratios can be achieved following input perturbations. Decreasing the excitatory input rate should lead to higher E/I ratios, while increasing it to lower E/I ratios.

The proposed rule suggests a new functional role of inhibitory plasticity, namely controlling E-to-E weight dynamics. Therefore, we extend previously studied roles of inhibitory plasticity, which include the stabilization of excitatory rates [44, 98], decorrelation of neuronal responses [120], preventing winner-take-all mechanisms in networks with multiple stable states [20] or generating differences among novel versus familiar stimuli [23]. Recent computational studies also include novel ways of inhibitory influence, like presynaptic inhibition via GABA spillover [121], an input-dependent inhibitory plasticity mechanism [122] and co-dependency of excitatory and inhibitory plasticity rules [123]. Our model includes a single type of inhibitory plasticity. Yet, recent studies have found that cortical circuits have abundance of different inhibitory interneuron types and that inhibitory plasticity depends on the inhibitory neuron type [75–78]. Our result on inhibitory population-dependent effects in gating receptive field formation suggests that subtype-specific plasticity rules might have non-trivial influences on the network, as some recent models have proposed [78, 124]. Furthermore, other homeostatic mechanisms will influence the stability of weight dynamics, E/I ratios and the effect different perturbations have on the network dynamics.

## Conclusion

Taken together, our study proposed a novel form of nonlinear inhibitory plasticity which can achieve stable firing rates and synaptic weights without the need for any additional homeostatic mechanisms. Moreover, our proposed plasticity is fast, and hence could provide a solution to the temporal paradox problem because it can counteract fast Hebbian excitatory plasticity. Functionally, our proposed inhibitory plasticity can establish and maintain a fixed E/I ratio for constant input rates at which the postsynaptic firing rate is exactly at the LTD/LTP threshold. For such postsynaptic firing rates, no synaptic plasticity is induced, i.e. plasticity is "off". Perturbing the postsynaptic firing rate, e.g. via disinhibition, can act as a gate,

turning plasticity "on". This enables the competition among input streams leading to receptive field formation in feedforward and recurrent circuits. Therefore, our nonlinear inhibitory plasticity mechanism provides a solution to the stability-flexibility challenge.

## Methods

### Rate-based model

We studied rate-based neurons to allow us to analytically investigate the dynamics of firing rates and synaptic weights in the model. In the feedforward motif (Fig 1A), we considered a network consisting of one excitatory postsynaptic neuron with a linear threshold transfer function and firing rate $v^E$, see Eq 1. The inhibitory neurons also follow a similar dynamics, see Eq 2. All parameters are given in Table 1. In the mean-field sense, the number of neurons can be traded-off with the rates or the synaptic weights, hence we assume $N^E = N^I = 1$ (Table 1).

### Rate-based plasticity

For the plasticity of E-to-E synaptic weights $w^{EE}$, we used a learning rule that depends nonlinearly on the postsynaptic rate $v^E$ (Fig 1B) [53–55]:

$$\tau_w^E \dot{w}^{EE} = \rho^E v^E (v^E - c_{post}^E). \tag{10}$$

Here, $\tau_w^E$ is the timescale of excitatory plasticity, which can be also thought of as the inverse of the learning rate, with correcting units $Hz^2$. This timescale is much longer than the timescale of the neuronal dynamics. The plasticity changes sign at the 'postsynaptic LTD/LTP threshold', $c_{post}^E$. During experimental induction of plasticity, low frequency stimulation (1,3 or 5 Hz) induces LTD, while high frequency stimulation (10–20 Hz) induces LTP [53]. Therefore, a natural value of the LTD/LTP threshold is between 5 and 10 Hz. We chose 1 Hz as the LTD/LTP threshold (Table 1), nonetheless, our findings will still hold with higher LTD/LTP thresholds.

For the plasticity of I-to-E synaptic weights $w^{EI}$, we used two learning rules. First, we used an inhibitory plasticity rule common in computational models [44, 56], which depends linearly on the postsynaptic rate $v^E$ (Fig 1D, $\dot{w}^{EI}$):

$$\tau_w^I \dot{w}^{EI} = v^I (v^E - c_{post}^I). \tag{11}$$

**Table 1. Parameter values for figures, ⋆ denotes that values are provided in the figure captions.**

| Sym. | Description | Fig 1 | Fig 2 | Fig 3 | Fig 4 | Fig 5 | Fig 6 | Fig 7B and 7C | S2 Fig | S3 Fig |
|---|---|---|---|---|---|---|---|---|---|---|
| $w_0^{EE}$ | Initial E-to-E weight | | ⋆ | | 1.5 | ⋆ | 1 | 0.7 | 1.5 | ⋆ |
| $w_0^{EI}$ | Initial I-to-E weight | | ⋆ | | 0.5 | ⋆ | 1 | 0.5 | | ⋆ |
| $w^{IE}$ | E-to-I weight | | | | 0.5 | | | | ⋆ | 0.5 |
| $\rho^E$ | Presyn. E rate (Hz) | | 2 | | ⋆ | 2 | | ⋆ | 2 | ⋆ |
| $\rho^I$ | Ext. E rate onto I neurons (Hz) | | | 0.5 | | | | ⋆ | | 0.5 |
| $N^E$ | Number of presyn. E neurons | | | | 1 | | | | | ⋆ |
| $N^I$ | Number of I neurons | | | | 1 | | | | | ⋆ |
| $\tau_{FR}^{E/I}$ | Time const. E/I rate dyn. (s) | | | | 0.01 | | | | | |
| $\tau_w^E$ | Timescale E plasticity ($Hz^2$) | | | | 1 | | | | 0.5 | 1 |
| $\tau_w^I$ | Timescale I plasticity (Hz or $Hz^2$) | | | | 0.2 | | | | 1 | 0.2 |
| $c_{post}^E$ | E postsyn. LTD/LTP thresh. (Hz) | 1 | ⋆ | | | | 1 | | | ⋆ |
| $c_{post}^I$ | I postsyn. LTD/LTP thresh. (Hz) | 1 | ⋆ | | | | 1 | | | ⋆ |

Here, $\tau_w^I$ denotes the timescale of inhibitory plasticity (or the inverse of the learning rate) with correcting units Hz, which again is much longer than the timescale of the neuronal dynamics. As for excitatory plasticity, inhibitory plasticity changes from LTD to LTP at the 'inhibitory postsynaptic LTD/LTP threshold', $c_{post}^I$, which sets the 'target rate' of the postsynaptic neuron [44]. In our paper, we proposed a novel inhibitory plasticity rule, which also depends nonlinearly on postsynaptic excitatory activity just like excitatory plasticity (Fig 2A):

$$\tau_w^I \dot{w}^{EI} = v^I v^E (v^E - c_{post}^I).$$ (12)

For both inhibitory plasticity rules, we assumed that the excitatory and inhibitory thresholds are matched ($c_{post}^E = c_{post}^I$) to prevent excitatory and inhibitory plasticity pushing the postsynaptic excitatory neuron towards two different firing rates. The exception for this was the dynamic mechanism for threshold matching in Fig 3 and S3 Fig.

**LTD/LTP plasticity thresholds.** As can be see in the equations for excitatory and inhibitory plasticity, the postsynaptic LTD/LTP thresholds, which determine the sign of plasticity as a function of postsynaptic excitatory activity, are fixed. However, in the main text we also introduce the concept of a presynaptic LTD/LTP threshold, defined as the presynaptic excitatory rate at which no synaptic plasticity is induced. We consider $v^E$ at steady state ($v^E = [N^E \rho^E w^{EE} - N^I v^I w^{EI}]_+$) and assume that the dynamics of the rates are in the region where the transfer function is linear. Therefore, we can drop the linear rectifier and solve for $\rho^E$ at which Eq 3 is zero. We derive the presynaptic LTD/LTP threshold as:

$$c_{pre}^E = \frac{c_{post} + N^I v^I w^{EI}}{N^E w^{EE}}.$$ (13)

**Stability analysis.** To investigate the stability of the weights, we first calculated the nullclines, where we assumed that the postsynaptic excitatory rate is at steady state $v^E = [N^E \rho^E w^{EE} - N^I v^I w^{EI}]_+$. By setting Eqs 10 and 12 to zero and dropping the linear rectifier, i.e. $v^E = N^E \rho^E w^{EE} - N^I v^I w^{EI}$, we can write:

$$\begin{aligned} w^{EI} &= \frac{N^E \rho^E w^{EE} - c_{post}^E}{N^I v^I}, \\ w^{EI} &= \frac{N^E \rho^E w^{EE} - c_{post}^I}{N^I v^I}. \end{aligned}$$ (14)

We see that the two equations are identical if $c_{post}^E = c_{post}^I$. Therefore, only for identical LTD/LTP thresholds ($c_{post}^E = c_{post}^I$) a line of fixed points emerges. The fixed points are $[w_*^{EE}, w_*^{EI}] = [x, (N^E \rho^E x - c_{post})/(N^I v^I)]$, where we require that $x \geq c_{post}/(N^E \rho^E)$ to avoid negative weights. To calculate the stability of the line of fixed points, we calculate the eigenvalues. We can rewrite Eqs 10 and 12, as:

$$\begin{aligned} \dot{w}^{EE} &= \frac{\rho^E}{\tau_w^E} \left( (N^E \rho^E w^{EE})^2 + (N^I v^I w^{EI})^2 - 2N^E N^I \rho^E v^I w^{EE} w^{EI} - N^E \rho^E w^{EE} c_{post} + N^I v^I w^{EI} c_{post} \right) = f \\ \dot{w}^{EI} &= \frac{v^I}{\tau_w^I} \left( (N^E \rho^E w^{EE})^2 + (N^I v^I w^{EI})^2 - 2N^E N^I \rho^E v^I w^{EE} w^{EI} - N^E \rho^E w^{EE} c_{post} + N^I v^I w^{EI} c_{post} \right) = g \end{aligned}$$ (15)

where we drop the linear rectifier by assuming that the dynamics of the rates are in the region where the transfer function is linear. We find that the entries of the Jacobian matrix

at the fixed points are:

$$J_* = \begin{pmatrix} \dfrac{\partial f}{\partial w^{EE}} & \dfrac{\partial f}{\partial w^{EI}} \\[3mm] \dfrac{\partial g}{\partial w^{EE}} & \dfrac{\partial g}{\partial w^{EI}} \end{pmatrix} = \begin{pmatrix} \dfrac{N^E (\rho^E)^2 c_{post}}{\tau_w^E} & -\dfrac{N^I \rho^E v^I c_{post}}{\tau_w^E} \\[3mm] \dfrac{N^E \rho^E v^I c_{post}}{\tau_w^I} & -\dfrac{N^I (v^I)^2 c_{post}}{\tau_w^I} \end{pmatrix}. \tag{16}$$

The trace of the Jacobian is $Tr(J_*) = \frac{N^E (\rho^E)^2 c_{post}}{\tau_w^E} - \frac{N^I (v^I)^2 c_{post}}{\tau_w^I}$ and the determinant is zero $Det(J_*)$ = 0, therefore we find that the eigenvalues are:

$$\lambda_{1,2} = \frac{1}{2}\left( Tr(J_*) \pm \sqrt{Tr(J_*)^2 - 4Det(J_*)} \right) = \begin{cases} Tr(J_*), \\ 0. \end{cases} \tag{17}$$

This means that if $Tr(J_*) < 0$, the system is stable. Reordering this condition leads to the stability condition derived in the main text as Eq 6. By reordering the terms in the nullclines given in Eq 14, we derive the line attractor equation as given in the main text in Eq 7.

The nonlinear excitatory and inhibitory plasticity rules have a fixed point when the postsynaptic excitatory firing rate is $v^E = 0$. Therefore, in the phase plane of $w^{EE}$ and $w^{EI}$ weights there is a region where the total inhibitory input is larger than the total excitatory input, $N^E \rho^E w^{EE} < N^I v^I w^{EI}$, resulting in no postsynaptic firing (Fig 2B, above gray line). The line equation separating the space with and without weight dynamics is:

$$w^{EI} = \frac{N^E \rho^E w^{EE}}{N^I v^I}. \tag{18}$$

In the case of the linear inhibitory plasticity rule, stability depends on the initial weights. The line which separates stable from unstable initial weights can be calculated by taking the ratio of Eqs 10 and 11 and equating that to the slope of the line attractor (Eq 7):

$$\frac{\dot{w}^{EI}}{\dot{w}^{EE}} = \frac{\tau_w^E v^I}{\tau_w^I \rho^E (N^E \rho^E w^{EE} - N^I v^I w^{EI})} = \frac{N^E \rho^E}{N^I v^I} \tag{19}$$

which leads to:

$$w^{EI} = \frac{N^E \rho^E}{N^I v^I} w^{EE} - \frac{v^I \tau_w^E}{N^E (\rho^E)^2 \tau_w^I}, \tag{20}$$

which is the equation of the dashed line in Fig 1E. The slope of the line attractor is the same for linear and nonlinear inhibitory plasticity.

In Eqs 13–20, the inhibitory firing rate can be replaced by its steady state value $v^I = N^E \rho^E w^{IE} + \rho^I$. For the stability condition (Eq 6) this leads to:

$$\frac{N^I (N^E \rho^E w^{IE} + \rho^I)^2}{\tau_w^I} > \frac{N^E (\rho^E)^2}{\tau_w^E}. \tag{21}$$

and for the line attractor (Eq 7) to:

$$w^{EI} = \frac{N^E \rho^E}{N^I (N^E \rho^E w^{IE} + \rho^I)} w^{EE} - \frac{c_{post}}{N^I (N^E \rho^E w^{IE} + \rho^I)}. \tag{22}$$

The perturbations of the presynaptic firing rate $\rho_{disr}^E$ in Fig 4 are defined as instantaneous and permanent increases or decreases from the initial presynaptic firing rate $\rho_{base}^E$.

## Dynamic threshold matching

The equations for the dynamics of the postsynaptic LTD/LTP thresholds in Fig 3D–3F and S3 Fig are:

$$
\begin{aligned}
\tau_{c_{post}^E} \dot{c}_{post}^E &= \dot{w}^{EE} \\
\tau_{c_{post}^I} \dot{c}_{post}^I &= -\dot{w}^{EI}
\end{aligned}
\tag{23}
$$

and therefore $c_{post}^E$ increases (decreases) if the postsynaptic neuron fires at $v^E > c_{post}^E$ ($v^E < c_{post}^E$) and $c_{post}^I$ decreases (increases) if the postsynaptic neuron fires at $v^E > c_{post}^I$ ($v^E < c_{post}^I$). The amount of increase or decrease of the postsynaptic thresholds is scaled by the amount of plasticity induction, and we used a timescale of $\tau_c^{E/I} = 2\ ms$, which is faster than the timescale of excitatory and inhibitory plasticity (Table 1). We point out that modifications in the LTD/LTP thresholds lead to changes in the induction of plasticity as well as the postsynaptic firing rate.

For two different initializations of the postsynaptic thresholds, $c_{post}^E < c_{post}^I$ and $c_{post}^E > c_{post}^I$, the synaptic weights, postsynaptic firing rate and postsynaptic threshold dynamics can be stabilized (Fig 3D–3F and S3A–S3C Fig). The same also holds when applying input perturbations (S3D–S3F Fig). For multiple input streams (S3G–S3I Fig), the dynamic postsynaptic LTD/LTP thresholds change based on the total excitatory (or inhibitory) weight change, leading to a non-local sliding mechanism which is independent of the input stream. A condition for the stabilization is that the weights do not reach their lower bounds at zero, because zero weights prevent plasticity and promote the continuous increase of LTD/LTP thresholds preventing firing rates from stabilizing.

## E/I ratio

We can calculate the E/I weight ratio $R^{E/I}$ in Eq 8 by rewriting Eq 14 and dividing one of the nullclines by $w^{EI}$. For large weights, or in mathematical terms for $w^{EI} \to \infty$, the E/I ratio becomes $\lim_{w^{EI} \to \infty} R^{E/I} = R_\infty^{E/I} = \frac{N^I v^I}{N^E \rho^E}$. This derivation is only valid for $N^I(N^E \rho^E w^{IE} + \rho^I) w^{EI} \gg c_{post}$. Therefore, the parameters of the input firing rates $\rho^E$ and $\rho^I$, the synaptic weights $w^{EI}$ and $w^{IE}$, as well as number of excitatory and inhibitory neurons $N^E$ and $N^I$ need to be chosen appropriately. This inequality is satisfied for the parameters in Fig 5 when the steady state synaptic weights $w^{EI}$ are sufficiently large (Table 1).

The existence of a fixed E/I ratio for constant input rates can be directly related to the line attractor. The line attractor (Eq 7) expresses the I-to-E weight $w^{EI}$ as a multiple of the E-to-E weight $w^{EE}$ minus the offset term $c_{post}/(N^I v^I)$. Therefore, the ratio of excitatory to inhibitory weight strengths, $R^{E/I}$ (Eq 8), can be expressed as the sum of two terms: one constant term equal to the slope of the line attractor, which is independent of the E-to-E and I-to-E weights, $w^{EE}$ and $w^{EI}$, and a second term, called an offset, which depends on $w^{EI}$. When this weight is sufficiently large, the offset term can be ignored, leading to an E/I ratio, $R_\infty^{E/I}$, independent from the E-to-E and I-to-E weights.

In the feedforward circuit (Fig 1A), we can write:

$$
R_\infty^{E/I} = \frac{N^I v^I}{N^E \rho^E} = \frac{N^I(\rho^I + w^{IE}\rho^E)}{N^E \rho^E} = \frac{N^I}{N^E}\left(\frac{\rho^I}{\rho^E} + w^{IE}\right).
\tag{24}
$$

Assuming that $N^E = N^I$, for larger excitatory input rate $\rho^E$ the E/I ratio reaches $R_\infty^{E/I} \approx w^{IE}$ (see

Fig 5A, where $w^{IE} = 0.5$). Therefore, the E/I ratio has a lower bound which depends on the strength of the connection from the excitatory to inhibitory population.

In Fig 5, we link our model to the experimental findings on how the interaction of excitatory and inhibitory plasticity can lead to fixed E/I ratios [37]. In [37], the authors induce plasticity with a spike-pairing protocol, in which pre-post spikes elicit excitatory LTP, while post-pre spikes elicit LTD. Inhibitory LTP was induced for short time differences between the pre- and postsynaptic spikes (independent of the order of the spikes) and inhibitory LTD for longer time differences of the spike pairs. Since in the experiments the presynaptic stimulation was done with a stimulation electrode, the excitatory and inhibitory inputs did not have to be functionally related. In the model, we randomly drew initial E-to-E and I-to-E weights and induced plasticity for a limited amount of time (100 ms) based on the rate-based plasticity rules (Eqs 10 and 12). We choose 100 ms so not all synaptic weights have reached the line attractor yet and so we can compare the E/I ratios reached in our model to those measured experimentally [37] which would most likely also not be in steady state.

The E/I balance can also be defined by the total excitatory input divided by the total inhibitory input onto the postsynaptic neuron:

$$R^{E/I}_{tot} = (N^E w^{EE} \rho^E)/(N^I w^{EI} v^I). \tag{25}$$

This leads to:

$$R^{E/I}_{tot} = (N^I(N^E \rho^E w^{IE} + \rho^I)w^{EI} + c_{post})/(N^I(N^E \rho^E w^{IE} + \rho^I)w^{EI}). \tag{26}$$

However, since we calculate the E/I balance at steady state, the total E/I balance is equal to the weight E/I balance multiplied by a constant, i.e.:

$$\tilde{R}^{E/I} = R^{E/I} N^E \rho^E/(N^I v^I). \tag{27}$$

Therefore, the results in Fig 5 also hold with this alternative E/I ratio definition.

## Noise and sinusoidal input

In Fig 6, we add a varying input either by modifying the presynaptic input rate $\rho^E$ or adding an additional term to the postsynaptic neuron (adding $\rho_{add}$ to Eq 1):

$$\tau^E_{FR} \dot{v}^E = -v^E + [N^E \rho^E w^{EE} - N^I v^I w^{EI} + \rho_{add}]_+. \tag{28}$$

In the case of postsynaptic noise (Fig 6A), $\rho_{add}$ is a normally distributed random variable with mean zero and standard deviation 0.01. In the case of additional sinusoidal input to the postsynaptic neuron (Fig 6B), $\rho_{add}(t) = 0.25 * \sin(0.01t)$. Recalculating the slope of the line attractor (Eq 7) based on Eq 28 leads to:

$$w^{EI} = \frac{N^E \rho^E}{N^I v^I} w^{EE} - \frac{c_{post} - \rho_{add}}{N^I v^I}, \tag{29}$$

meaning that $\rho_{add}$ only affects the intersection, but not the slope of the line attractor. We note that the line attractor is calculated at steady state firing rates, meaning that the line attractor will actually never be reached by a varying input.

In the case of presynaptic noise (Fig 6D), we add a normally distributed random variable with mean zero and standard deviation 0.3 to the presynaptic firing rate $\rho^E$. For the sinusoidal input (Fig 6E), we chose $\rho^E(t) = 2 + 0.5 * \sin(0.01t)$.

## Gating of receptive field formation and recurrent clustering

Here, we explore a feedforward network with multiple inputs and two inhibitory neuron populations (Fig 7C). To form receptive fields, we provide a random patterned input to the network. An input pattern is defined by a high firing rate of 4 Hz at a subset of four excitatory input neurons for a time of 100 ms. In Eqs 1 and 2, this is reflected by a subset of the $N^E$ inputs having $\rho_m^E = 4$ Hz, where $m$ corresponds to the presynaptic neurons being part of the respective input pattern. After a time of 100 ms, a new subset of four excitatory neurons fire at high firing rates. We then disinhibit the postsynaptic neurons by inhibiting either the total unspecific or specific inhibitory populations for 60 s by inducing an inhibitory input of 2 onto the respective inhibitory neuron population (we set $\rho_{spec}^I = -2$ or $\rho_{unsp}^I = -2$). Disinhibition needs to be applied for a sufficiently long time to ensure that inhibitory plasticity can induce competition and form receptive fields. We model the release of disinhibition for the specific inhibitory population as slow and gradual over a time course of 100 s to avoid complete silencing of the postsynaptic excitatory neurons. We also note that here we used instantaneous integrators, i.e. $\tau_{FR}^E = \tau_{FR}^I = dt$ (Table 2), because we only wanted to focus on the interaction of excitatory and inhibitory plasticity in the model, though see [125].

For the recurrent circuit, we connected recurrently 30 postsynaptic neurons with feedforward circuits with specific and unspecific inhibition as described above (see also Fig 7D and 7E). In addition to feedforward excitatory and inhibitory weights, also recurrent excitatory weights were plastic based on the plasticity mechanism of Eq 10. We allowed the input patterns to each of the recurrent excitatory neuron to be correlated. Initial recurrent excitatory weights were randomly drawn from the interval [0,0.18]. We calculated the mean weight per input pattern and chose the maximum of those to be the input to which the neurons formed a receptive field. The clustering graph in Fig 7F (left) was done with the digraph function in Matlab where the distance between neurons is only used to visualize clusters of neurons with similar tuning.

**Table 2. Parameter values for Fig 7E and 7F.**

| Symbol | Description | Fig 7E | Fig 7F |
|:---:|:---:|:---:|:---:|
| $w_0^{EE}$ | Initial E-to-E weight | 0.03 | [0,0.18] |
| $w_{spec,0}^{EI}$ | Initial specific I-to-E weight | 0.01 | |
| $w_{unsp,0}^{EI}$ | Initial unspecific I-to-E weight | 0.01 | |
| $w_{spec}^{IE}$ | Specific E-to-I weight (fixed) | 0.2 | 0.002 |
| $w_{unsp}^{IE}$ | Unspecific E-to-I weight (fixed) | 0.02 | 0.001 |
| $\rho^E$ | Presynaptic E rate (Hz) | 1 | |
| $\rho_{spec}^I$ | External E rate onto specific I neurons (Hz) | 0 | |
| $\rho_{unsp}^I$ | External E rate onto unspecific I neurons | 0 | |
| $N^E$ | Number of presyn. E neurons (Hz) | 40 | |
| $N_{spec}^I$ | Number of specific I neurons | 20 | |
| $N_{unsp}^I$ | Number of unspecific I neurons | 20 | |
| $\tau_{FR}^E$ | Timescale for E neuron model (s) | 0.0001 | |
| $\tau_{FR}^I$ | Timescale for I neuron model (s) | 0.0001 | |
| $\tau_w^E$ | Timescale for E plasticity (Hz$^2$) | 1 | |
| $\tau_w^I$ | Timescale for I plasticity (Hz$^2$) | 0.2 | |
| $c_{post}^E$ | E postsyn. LTD/LTP threshold (Hz) | 1 | |
| $c_{post}^I$ | I postsyn. LTD/LTP threshold (Hz) | 1 | |

The simulations were performed using Matlab programming language. Euler integration was implemented using a time step of 0.1. Code implementing our model is available here: https://github.com/comp-neural-circuits/Nonlinear-inhibitory-plasticity.

## Supporting information

**S1 Fig. Plasticity of excitatory-to-excitatory synapses as a function of presynaptic and post-synaptic firing rates.** Excitatory plasticity $\dot{w}^{EE}$ (Eq 3) is normalized to the maximum value of long-term potentiation (1) and the maximum value of long-term depression (−1), respectively. (EPS)

**S2 Fig. Feedback inhibitory motif leads to additional stability. A**. Schematic of the feedback inhibitory motif. The inhibitory population receives input from the presynaptic excitatory population with weight strength $w_{FF}^{IE}$ and the excitatory postsynaptic neuron with weight strength $w_{FB}^{IE}$. **B**. Plasticity of E-to-E ($\dot{w}^{EE}$, blue) and I-to-E ($\dot{w}^{EI}$, red) weights as a function of the postsynaptic rate $v^E$. The excitatory and inhibitory LTD/LTP thresholds are identical ($c_{post}^E = c_{post}^I$). **C**. E-to-E ($w^{EE}$, blue) and I-to-E ($w^{EI}$, red) and rate dynamics of the postsynaptic (gray line) and the inhibitory population (gray dashed line) as a function of time. **D**. Stability of weight dynamics as a function of the excitatory-to-inhibitory weights $w_{FB}^{IE}$ and $w_{FF}^{IE}$. Star indicates the values shown in panel C. (EPS)

**S3 Fig. Dynamic matching of the excitatory and inhibitory postsynaptic LTD/LTP thresholds and networks response to input perturbations. A**. Postsynaptic LTD/LTP thresholds $c_{post}^E$ and $c_{post}^I$ shift dynamically depending on the recent postsynaptic rate $v^E$. For lower postsynaptic rate than the excitatory postsynaptic LTD/LTP threshold ($v^E < c_{post}^E$), $c_{post}^E$ decreases, and for $v^E > c_{post}^E$, $c_{post}^E$ increases. For higher postsynaptic rate than the inhibitory postsynaptic LTD/LTP threshold ($v^E > c_{post}^I$), $c_{post}^I$ decreases, and for $v^E < c_{post}^I$, $c_{post}^I$ increases (see Methods). **B**. Evolution of excitatory ($c_{post}^E$, blue) or inhibitory ($c_{post}^I$, red) postsynaptic LTD/LTP thresholds for initial conditions $c_{post,0}^E = 1.3$, $c_{post,0}^I = 0.7$. **C**. Excitatory ($w^{EE}$, blue) and inhibitory ($w^{EI}$, red) weight dynamics and postsynaptic rate dynamics ($v^E$, gray) for the initial condition $c_{post,0}^E = 1.3$, $c_{post,0}^I = 0.7$. **D**. Effect of increasing (solid lines, $\rho_{disr}^E = 2.5$) or decreasing (dashed lines, $\rho_{disr}^E = 1.5$) excitatory input rates from a baseline of $\rho_{base}^E = 2$ on excitatory (blue) and inhibitory (red) firing rates. **E**. Same as D but for the $c_{post}^E$ and $c_{post}^I$ weights. **F**. Same as D but for the $w^{EE}$ and $w^{EI}$ weights. **G**. Plasticity curve of E-to-E weights for input 1 or 2 ($\dot{w}_{1,2}^{EE}$) as a function of the presynaptic excitatory rate $\rho^E$ for different input-specific perturbations $\rho_{disr,1,2}^E$. **H**. Evolution of excitatory ($c_{post}^E$, blue) or inhibitory ($c_{post}^I$, red) postsynaptic LTD/LTP thresholds for the case in G. **I**. Excitatory ($w^{EE}$, blue) and inhibitory ($w^{EI}$, red) weight dynamics for the case in G. Compare A-C to Fig 3, D-F to Fig 4 and G-I to Fig 7B. (EPS)

**S4 Fig. Performance of the nonlinear inhibitory plasticity rule under varying postsynaptic firing rate with dynamic excitatory and inhibitory LTD/LTP threshold matching. A**. Adding noise to the postsynaptic firing rate. Top: E-to-E ($w^{EE}$, blue) and I-to-E ($w^{EI}$, red) as a function of time. Middle: Excitatory ($c_{post}^E$, blue) and inhibitory ($c_{post}^I$, red) postsynaptic LTD/LTP threshold as a function of time. Bottom: Postsynaptic rate dynamics ($v^E$, gray) as a function of time. **B**. Same as A but after adding a sinusiodal input to the postsynaptic firing rate. (EPS)

## Acknowledgments

We thank all members of the 'Computation in Neural Circuits' group, and specifically Yue Kris Wu, for useful discussions and comments on the manuscript.

## Author Contributions

**Conceptualization:** Christoph Miehl, Julijana Gjorgjieva.

**Formal analysis:** Christoph Miehl.

**Funding acquisition:** Julijana Gjorgjieva.

**Investigation:** Christoph Miehl.

**Project administration:** Julijana Gjorgjieva.

**Software:** Christoph Miehl.

**Supervision:** Julijana Gjorgjieva.

**Visualization:** Christoph Miehl.

**Writing – original draft:** Christoph Miehl, Julijana Gjorgjieva.

**Writing – review & editing:** Christoph Miehl, Julijana Gjorgjieva.

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
