## [Decision Letter · Decision Letter 0]

8 Jun 2022

Dear Dr Gjorgjieva,

Thank you very much for submitting your manuscript "Stability and learning in excitatory synapses by nonlinear inhibitory plasticity" for consideration at PLOS Computational Biology.

As with all papers reviewed by the journal, your manuscript was reviewed by members of the editorial board and by several independent reviewers. In light of the reviews (below this email), we would like to invite the resubmission of a significantly-revised version that takes into account the reviewers' comments.

We cannot make any decision about publication until we have seen the revised manuscript and your response to the reviewers' comments. Your revised manuscript is also likely to be sent to reviewers for further evaluation.

Sincerely,

Hugues Berry

Associate Editor

PLOS Computational Biology

Kim Blackwell

Deputy Editor

PLOS Computational Biology

Reviewer's Responses to Questions

**Comments to the Authors:**

Reviewer #1: This is an interesting paper addressing an important topic of stability of operation of neuronal microcircuits in face of on-going synaptic plasticity. The authors propose a novel mechanism that may help to stabilize activity in a model of canonical feedforward excitatory/inhibitory microcircuit. The authors demonstrate that the proposed mechanism can also regulate, by affecting the rate of postsynaptic neuron, plasticity at excitatory synapses and thus serve as a mechanism of metaplasticity.

The paper will make valuable contribution towards mechanistic understanding of plasticity in neuronal microcircuits.

There are several points, however, that have to be addressed.

1. In the Abstract (may be also in the title), it should be clearly indicated that this is a purely modelling study.

2. Introduction

L10-18: experimentally supported homeostatic mechanisms: synaptic scaling, heterosynaptic plasticity, plasticity of intrinsic excitability

... Part of 32

... the challenge is that the experimentally measured timescales of homeostatic mechanisms 33

... are too slow to stabilize the Hebbian runaway dynamics in computational models, 34

... sometimes referred to as the ‘temporal paradox’ of homeostasis [19–21].

similarly in the Discussion (lines 372-374).

The problem of fundamentally different timescales (seconds vs many hours and days) is true ONLY for synaptic scaling, which thus fundamentally cannot counteract positive feedback of Hebbian-type plasticity but nevertheless referred too most extensively.

Heterosynaptic plasticity is induced on exact same time scale as Hebbian-type plasticity and intrinsic plasticity – on a similar time scale. So the 'temporal paradox' applies only to synaptic scaling, but not to heterosynaptic plasticity or intrinsic plasticity. Please explain this.

3. Introduction

... However, how exactly synaptic plasticity and homeostatic mechanisms interact to 31

... control synaptic strengths, and yet enable learning, is still unresolved [16–18].

While indeed there is no consensus on this issue, there are at least attempts to do so. The ability of experimentally-observed heterosynaptic plasticity to counteract destabilizing effects of STDP on synaptic weights and activity, while allowing learning was demonstrated in model neurons and simple networks (Chen et al 2013; Volgushev et al 2016). In fact, even evidence for the relevance of heterosynaptic plasticity for behavior learning was presented (Chasse et al 2020). Also 'Mexican hat' profile of homo- and hetero-synaptic plasticity can provide a mechanism for balancing synaptic changes (White et al 1990; Royer, Pare 2003).

4. Excitatory plasticity rule used in this study is highly unbalanced, with only weak depression and strongly dominating potentiation. Also rate for LTD/LTP threshold is low (1 Hz). However in physiological experiments low frequency stimulation with 1,3 or 5 Hz rather induces LTD, and LTP is induced at 10-20 Hz or higher. So frequency threshold in conventional BCM rule would be around 10 Hz. I see that there is no direct translation of rates from rate models to conventional BCM/STDP/tetanus induction protocols, and 1 Hz threshold might be 'typical' for rate models, but it would helpful to discuss this issue.

5. Throughout the results and figures, please always indicate postsynaptic firing rates since it is critical for understanding in which region of plasticity curve the model is operating. Yes, firing rate is indicated in most cases, please add it where it's missing.

6. Fig 4 – add a plot of plasticity rules used in this simulation.

7.

... their potentiation (Fig 4C). The response to these perturbations is consistent with 241

... previous experimental results. For example, it has been shown that input perturbations 242

... via sensory deprivation decrease inhibitory activity [49–51].

Deprivation effects are developed on timescale of hours/days; in the model (Fig. 4C) within less than a second. I would say you are introducing here an 'inverse temporal paradox' trying to say that a process completed within a second could underlie day-long gradual changes. Please explain/discuss.

8. Same problem as above, very different time scales, holds for prediction of the steady state of E/I ratio (from line 269). In the model, steady state is reached within few seconds; this could be time scale of sensory adaptation, but not of say deprivation-induced changes. Please explain/discuss.

9.

... Decreasing the excitatory input rate decreases the excitatory presynaptic 253

... LTD/LTP threshold, hence

This sounds counter intuitive – decreasing presynaptic rate might make it more difficult to reach fixed postsynaptic threshold rate. Or, because of inhibitory-dominated circuit decrease of presynaptic rate would have strong disinhibiting effect which 'overrules' the decrease of presynaptic input? Or am I missing something? Please explain.

Also, in Fig 6B it's the other way round –presynaptic LTD/LTP threshold increases for lower presynaptic rate, and decreases for higher. This is more intuitive.

Please explain the difference between Fig 4E and 6B.

10.

... We found that disinhibiting the 322

... unspecific inhibitory population does not selectively potentiate E-to-E weights, and 323

... hence does not generate competition among the different inputs. In contrast, 324

... disinhibiting all ten specific inhibitory populations strongly increases the E-to-E weights 325

... corresponding to only a subset of inputs, a process also called receptive field formation 326

... (Fig 6E).

Please add postsynaptic firing rate trace to Fig 6E.

In the first case, please explain why changes of the postsynaptic firing rate did not induce plasticity. Were changes of firing rate after nonspecific disinhibition same as after the specific?

In the second case, inputs were probably disinhibited non-equally? Please provide details.

And, if disinhibition was same for all inputs, please explain why changes of weights were heterogeneous (or random?).

11.

Fig 6F please explain the connectivity. Were in this simulation the recurrently connected neurons added in place of the big triangle in the middle of 6D? In the scheme, there are multiple presynaptic neurons in each input, but only one postsynaptic.

In 6F – do location of individual neurons and distance between them have any meaning? Or is it just to make the whole thing look like a brain (which is OK too), but please explain is there is a meaning.

Reviewer #2: Review was uploaded as an attachment.

Reviewer #3: The stability of neural activity requires a certain degree of the robustness of synaptic configuration in neural networks, whereas learning requires flexibility in remodeling the existing synaptic structures. Therefore, stable neural dynamics and learning new experiences have conflicting demands. The authors studied how STDP and homeostatic plasticity interact during the development of neural circuits to organize an excitation-inhibition balance that simultaneously fulfills these demands. Their solution is based on inhibitory plasticity with a nonlinear learning rule. The authors proposed a class of inhibitory plasticity rules at I-to-E synapses to cancel out excitatory plasticity's inherently positive feedback effects.

Although some results are of potential interest, I have several concerns about the validity of the main results. I cannot exclude the possibility that I have missed some crucial points. However, I feel that the present analysis is not sufficient to confirm the central claim of this study. For the acceptance of this manuscript, the following concerns should be clarified.

Major comments:

1. The previous linear inhibitory plasticity requires a fine-tuning of the target firing rates of excitatory and inhibitory plasticity rules: c_post^E = c_post^I. In the nonlinear inhibitory plasticity proposed in this manuscript, this condition was relaxed by introducing dynamical modulations of the target firing rates. However, I have difficulty understanding the rules of these modulations. The modulation rules seem to work if vE>c_post^E, c_post^I, and c_post^E < c_post^I at time t=0. In this case, c_post^E is increased, and c_post^I is decreased in time; hence c_post^E = c_post^I is eventually achieved. However, when vE<c_post^e, ve="">c_post^E, vE<c_post^i, and="" c_post="">

2. Related to the above comment, I wonder whether the initial state should still satisfy the condition c_post^I > c_post^E even in the dynamical tuning mechanism for c_post^E and c_post^I. If so, this significantly limits the validity of the proposed non-linear inhibitory plasticity rule. Such cases are not studied in Fig. 3. The tuning procedure should be described mathematically more explicitly in the Methods.

3. For the reasons mentioned in comments 1 and 2, the perturbation analysis in Fig. 4 should be conducted with the dynamical tuning mechanism of the threshold values without assuming the equality c_post^I = c_post^E. These simulations, however, will not be necessary if the authors prove the threshold matching for arbitrary initial values of the variables and parameters.

4. In the paragraph on ll. 402-420, the authors discussed the possible relationship between their homeostatic mechanism via inhibitory plasticity rule and the conventional BCM theory, assuming the existence of slow and fast homeostatic mechanisms. The authors suggested that the slow mechanism depends on the cell's intrinsic excitability or synaptic scaling while the fast mechanism on disinhibition and inhibitory plasticity (the cases studied in their model). Please cite references, if any, which give supportive evidence for the different time scales in intrinsic excitability/ synaptic scaling and disinhibition/inhibitory plasticity.

5. The authors vaguely argued the consistency between the computational results and experimental results in several places. For instance, on ll. 441-442, the authors mentioned, "We found that the new nonlinear inhibitory plasticity rule achieves an E/I ratio set point (Fig. 5) in agreement with experimental data [26]. However, in what sense is the theoretically obtained set point consistent with the experimental observations? Although I do not list all these places, I want to see more precise statements.

Minor comments:

6. In the legend of Fig. 2C, please explain the meanings of solid and dashed lines although we can guess the meanings.

7. On lines 494-504, the authors suggested that specific and nonspecific inhibitory neurons in the model correspond to SOM+ interneurons and PV+ interneurons, respectively. However, I doubt whether this assumption is biologically plausible from the viewpoint of neuroanatomy. The model's network structure suggests that the specific inhibitory neurons, which target specific excitatory neurons, are the dominant inhibitory neuron type, i.e., PV+ interneuron in local cortical circuits.</c_post^i,></c_post^e,>

**Have the authors made all data and (if applicable) computational code underlying the findings in their manuscript fully available?**

Reviewer #1: Yes

Reviewer #2: Yes

Reviewer #3: Yes

PLOS authors have the option to publish the peer review history of their article (what does this mean?). If published, this will include your full peer review and any attached files.

Reviewer #1: **Yes: **Maxim Volgushev

Reviewer #2: No

Reviewer #3: No
---

## [Decision Letter · Decision Letter 1]

17 Oct 2022

Dear Dr Gjorgjieva,

Thank you very much for submitting your manuscript "Stability and learning in excitatory synapses by nonlinear inhibitory plasticity" for consideration at PLOS Computational Biology. As with all papers reviewed by the journal, your manuscript was reviewed by members of the editorial board and by several independent reviewers. The reviewers appreciated the attention to an important topic. Based on the reviews, we are likely to accept this manuscript for publication, providing that you modify the manuscript according to the review recommendations.

Pplease be sure to take into accounts the remaining points raised by reviewer#2, in particular the point pertaining to the E/I ration set point. 

Sincerely,

Hugues Berry

Academic Editor

PLOS Computational Biology

Kim Blackwell

Section Editor

PLOS Computational Biology

Reviewer's Responses to Questions

**Comments to the Authors:**

Reviewer #1: All my comments are addressed adequately.

Reviewer #2: The review is uploaded as an attachment.

Reviewer #3: Supplementary Fig.3 and other revisions by the authors clarified all my concerns. I have no further comments on the revised manuscript, which I think is acceptable for publication in PLoS Computational Biology.

**Have the authors made all data and (if applicable) computational code underlying the findings in their manuscript fully available?**

Reviewer #1: None

Reviewer #2: **No: **Code for Figs 1 to 6 is provided in their Github link, but not for Fig 7.

Reviewer #3: Yes

PLOS authors have the option to publish the peer review history of their article (what does this mean?). If published, this will include your full peer review and any attached files.

Reviewer #1: **Yes: **Maxim Volgushev

Reviewer #2: No

Reviewer #3: No

Figure Files:

Data Requirements:

Reproducibility:

References:

---

## [Decision Letter · Decision Letter 2]

25 Oct 2022

Dear Dr Gjorgjieva,

We are pleased to inform you that your manuscript 'Stability and learning in excitatory synapses by nonlinear inhibitory plasticity' has been provisionally accepted for publication in PLOS Computational Biology.

Best regards,

Hugues Berry

Academic Editor

PLOS Computational Biology

Kim Blackwell

Section Editor

PLOS Computational Biology

Reviewer's Responses to Questions

**Comments to the Authors:**

Reviewer #2: The authors have addressed all of my concerns, and I'm satisfied with the revisions.

I appreciate the authors' effort in sharing the code to generate the manuscript's plots. In this regard, I would like to suggest renaming the files in their github repository to match the manuscript's latest version. Additionally, I could not find the code for the new Fig 6 in the github repository, and I think it would be important to share this code for completeness.

Best wishes.

**Have the authors made all data and (if applicable) computational code underlying the findings in their manuscript fully available?**

Reviewer #2: **No: **Fig 6 of the latest submission is not yet shared in the github repository.

PLOS authors have the option to publish the peer review history of their article (what does this mean?). If published, this will include your full peer review and any attached files.

Reviewer #2: No

---

## [Editor Report · Acceptance letter]

7 Nov 2022

PCOMPBIOL-D-22-00534R2 

Stability and learning in excitatory synapses by nonlinear inhibitory plasticity

Dear Dr Gjorgjieva,

I am pleased to inform you that your manuscript has been formally accepted for publication in PLOS Computational Biology. Your manuscript is now with our production department and you will be notified of the publication date in due course.

With kind regards,

Zsofia Freund
